# Interplay of kinetochores and catalysts drives rapid assembly of the mitotic checkpoint complex

Suruchi Sethi[1,3,5], Sabrina Ghetti [1,5], Verena Cmentowski [1], Teresa Benedetta Guerriere[1], Patricia Stege [1], Valentina Piano [1,4] & Andrea Musacchio [1,2] ✉

The spindle assembly checkpoint (SAC) ensures mitotic exit occurs only after sister chromatid biorientation, but how this coordination is mechanistically achieved remains unclear. Kinetochores, the megadalton complexes linking chromosomes to spindle microtubules, contribute to SAC signaling. However, whether they act solely as docking platforms or actively promote the co-orientation of SAC catalysts such as MAD1:MAD2 and BUB1:BUB3 remains unresolved. Here, we reconstitute kinetochores and SAC signaling in vitro to address this question. We engineer recombinant kinetochore particles that recruit core SAC components and trigger checkpoint signaling upon Rapamycin induction, and test their function using a panel of targeted mutants. At approximately physiological concentrations of SAC proteins, kinetochores are essential for efficient mitotic checkpoint complex (MCC) assembly, the key effector of SAC signaling. Our results suggest that kinetochores serve not only as structural hubs but also as catalytic platforms that concentrate and spatially organize SAC components to accelerate MCC formation and ensure timely checkpoint activation.

In eukaryotes, faithful chromosome segregation during mitosis depends on the formation of bioriented attachments between the sister chromatids and the mitotic spindle. The attachment of sister chromatids to microtubules is mediated by kinetochores, multisubunit microtubule-binding machines built on specialized chromosome loci called centromeres[1,2]. The microtubule attachment process is monitored by specialized machinery, also residing at mitotic kinetochores, that corrects improper kinetochore-microtubule attachments while coordinating the progression of chromosome biorientation with mitotic exit. The feedback mechanism responsible for this coordination is named the spindle assembly checkpoint (SAC)[3–6]. It imposes a delay to mitotic exit in the presence of conditions that interfere with the completion of biorientation.

The SAC effector, known as the mitotic checkpoint complex (MCC), is a complex of the four proteins CDC20, MAD2, BUBR1, and BUB3[7–9]. It assembles at unattached kinetochores and inhibits cell cycle progression by binding and inhibiting the anaphase promoting complex or cyclosome (APC/C) (Fig. 1A)[7,10–15]. How kinetochores control the rate of MCC assembly is a question of great interest. In vitro, MCC assembly from its subunits occurs spontaneously at concentrations resembling cellular concentrations, but the rate of spontaneous MCC assembly is very slow[16–18]. A topological rearrangement of the HORMA domain protein MAD2 is rate-limiting. MAD2 adopts a metastable open topology (O-MAD2) that converts into the more stable closed topology (C-MAD2). This conversion occurs simultaneously with the binding to a "closure

[1]Department of Mechanistic Cell Biology, Max Planck Institute of Molecular Physiology, Otto-Hahn-Straße 11, 44227 Dortmund, Germany. [2]Centre for Medical Biotechnology, Faculty of Biology, University Duisburg-Essen, Essen, Germany. [3]Present address: Eradigm Consulting, 6-7 St Cross St, London, EC1N 8UB, UK. [4]Present address: Institute of Human Genetics, Center of Molecular Medicine Cologne (CMMC), University of Cologne, Robert-Koch Str. 21 50931, Cologne, Germany. [5]These authors contributed equally: Suruchi Sethi, Sabrina Ghetti. ✉e-mail: andrea.musacchio@mpi-dortmund.mpg.de

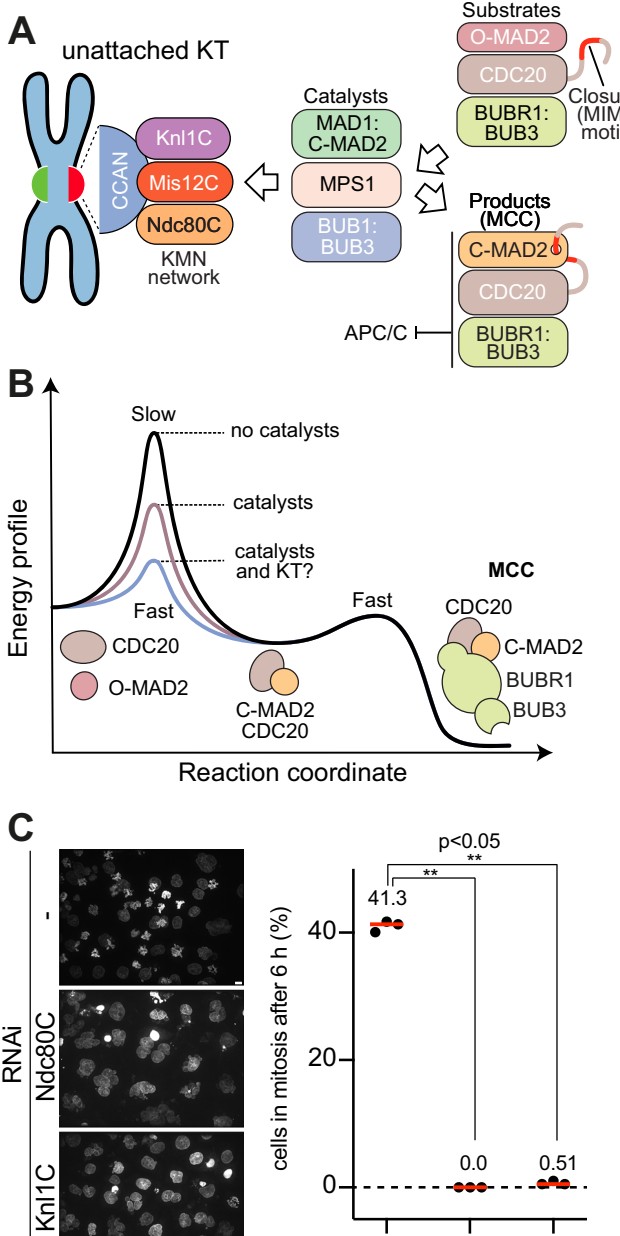

**Fig. 1 | The outer kinetochore and catalysts promote SAC signaling. A** Scheme of catalytic assembly of MCC. Catalysts for MCC assembly are recruited to the outer kinetochore, consisting of Knl1C, Mis12C and Ndc80C. The catalysts recruit the MCC substrates and turn them into the product, the MCC, which diffuses away and binds and inhibits the APC/C. **B** Energy profile of MCC formation. Rate-limiting for MCC assembly is the association of CDC20:C-MAD2 from CDC20 and O-MAD2. It is spontaneous but slow due to the high activation energy. A bipartite assembly comprising BUB1:BUB3 and the MAD1:C-MAD2 core complex, activated by MPS1, catalyzes this association by lowering the energy barrier. MCC further incorporates BUBR1:BUB3 in a rapid spontaneous reaction. The hypothesis is that kinetochores (KT) further lower the activation energy. **C** Analysis of mitotic checkpoint functionality in cells depleted of the Ndc80C or the Knl1C. HeLa cells were arrested in the G2 phase of the cell cycle, released into mitosis in the presence of 33 nM Nocodazole and imaged 6 h after release. The percentage of mitotic cells is shown. Collectively, 678, 270, and 588 cells were analyzed for the control, Ndc80C RNAi, and Knl1C RNAi conditions, respectively, in three repeats of the experiment. The red line represents the median. Statistical analysis was performed with a two-sided non-parametric Mann-Whitney test comparing two unpaired groups; p-values were as follows: CTRL-Ndc80C RNAi p-value 0.0013 and CTRL-Knl1C RNAi p-value 0.0079. Scale bar = 10 μm.

motif" (also known as MIM, for MAD2-interacting motif) of CDC20[16,17,19–25].

While very slow in vitro, this conversion occurs rapidly at kinetochores, indicating that it may be catalyzed (Fig. 1B)[26]. Work in vitro and in vivo has now demonstrated that a scaffold of additional SAC proteins at the kinetochore acts as a recruiting platform for O-MAD2 and CDC20 and catalyzes their combination[16,18,23,25]. This scaffold comprises the MAD1:C-MAD2 core complex, the BUB1:BUB3 complex, and the MPS1 kinase (Fig. 1A). MAD1, a coiled-coil protein unrelated to CDC20 except for the presence of a MIM motif, holds C-MAD2 in a very stable 2:2 tetramer, the MAD1:C-MAD2 core complex[20,27]. The C-MAD2 molecules in the MAD1:MAD2 core complex recruit O-MAD2 from the cytosol through an asymmetric "conformational dimerization"[22,28]. BUB1:BUB3 interacts with MAD1:C-MAD2, and together these proteins further recruit CDC20 to allow its rapid combination with O-MAD2[16,23,29–36]. Swiveling of the C-terminal region of MAD1 is required for this conversion[32,33,37,38]. These processes are collectively orchestrated by MPS1 kinase, which phosphorylates BUB1 to promote its interaction with MAD1, activates MAD1 through phosphorylation of its C-terminal region, and, in metazoans, promotes assembly of a specialized structure, the kinetochore corona, that cements these interactions before being dissolved by microtubule binding[16,30,31,34–36,39–41]. A set of SAC catalysts comprising MAD1:MAD2, BUB1:BUB3, and MPS1 is sufficient to promote a very significant acceleration of MCC assembly in an in vitro reconstituted system even in the absence of kinetochores[16,23].

A single unattached kinetochore supports enough MCC assembly to maintain a mitotic delay[26,42]. Thus, we suspect that the catalytic efficiency of MCC assembly at kinetochores may be significantly higher than the one measured in vitro with isolated components. How kinetochores contribute to SAC signaling, however, remains unclear. All SAC components are recruited to the outer kinetochore, a 10-subunit assembly consisting of the KNL1, MIS12, and NDC80 subcomplexes, herewith respectively designated as Knl1C, Mis12C, and Ndc80C[2,6]. Specifically, MPS1 kinase becomes recruited through an interaction with Ndc80C, which also serves as a microtubule-attachment platform through the NDC80 subunit (also known as HEC1)[43–46]. Instead, BUB1:BUB3 becomes recruited by so-called MELT motifs in the KNL1 protein, after their phosphorylation by MPS1 kinase and recognition by BUB3, a phospho-aminoacid adapter[47–50]. Precisely how MAD1:MAD2 becomes recruited to kinetochores has remained partly elusive. The Ndc80C is required[51], possibly because of its role in recruiting MPS1, while BUB1 and the corona may provide direct binding sites[30,31,34,37,52–55].

How kinetochores influence MCC assembly is less clear. Depletion of KMN subunits has resulted in a range of different responses, likely because the penetrance of the depletion strongly influences the outcome, with partial depletions being compatible with a strong mitotic arrest, and complete depletion causing checkpoint override[51,56–59]. Collectively, however, the current evidence is consistent with the notion that kinetochores are essential for SAC signaling. In addition to providing binding sites for the SAC catalysts, kinetochores may be expected to promote their productive interactions by orienting them favorably for their catalytic function on the MCC subunits. Furthermore, kinetochores are expected to couple microtubule binding to SAC suppression, a process whose molecular details are only beginning to be uncovered[3,5]. Here, we make additional inroads into the dissection of these complex questions by combining biochemical in vitro reconstitutions of the human SAC and kinetochores. We demonstrate that kinetochores accelerate MCC accumulation also at high catalyst concentration, but become indispensable for rapid accumulation of MCC at physiological catalyst concentration. Our observations are consistent with a model in which both the concentration of SAC catalysts and their co-orientation promote efficient catalytic accumulation of MCC.

## Results and Discussion

### Kinetochores are required for the SAC response

To assess the role of kinetochores in the SAC response, we depleted the outer kinetochore complexes Ndc80C or Knl1C by RNAi using pools of siRNAs (directed against NDC80, SPC24, and SPC25 for the Ndc80C, and against KNL1 and ZWINT for the Knl1C; see "Methods"). We then monitored the accumulation of mitotic cells 6 h after their release from a G2 arrest in the presence of the microtubule poison Nocodazole, a potent SAC activator, counting the fraction of cells that had remained in mitosis with an active SAC (Fig. 1C). More than 40% of cells in the control condition were mitotic, as revealed by condensation of their chromosomes. Cells depleted of the outer kinetochore complexes, on the other hand, had all exited mitosis prematurely and had re-entered interphase. Thus, both Knl1C and Ndc80C are necessary to sustain the SAC. The SAC-promoting function of unattached kinetochores likely reflects their ability to recruit SAC proteins and to orient them reciprocally for catalytic MCC production. We therefore set out to shed light on this mechanism through biochemical reconstitution, harnessing our previous reconstitutions of SAC signaling and kinetochore organization with recombinant proteins[16,23,60,61].

### Engineering an inducible functional SAC scaffold: MPS1 and MAD1

A crucial challenge towards achieving a functional biochemical reconstitution of kinetochores in SAC signaling is that the reconstituted kinetochore particles ought to recruit all SAC proteins. This goal, however, has not yet been fully achieved in vitro. A first challenge concerns MPS1, an apical kinase in SAC pathway. MPS1 contributes to SAC signaling in multiple ways. For instance, it phosphorylates the Met-Glu-Leu-Thr (MELT) motifs on KNL1 to recruit the BUB1:BUB3 complex[47–50]. It also phosphorylates, after CDK1 priming phosphorylation, Thr461 of BUB1, located in the conserved motif 1 (CM1, residues 458-476), to promote the interaction of BUB1 with a conserved Arg-Leu-Lys (RLK) motif in the C-terminal coiled-coil region of MAD1[30,31,34–36]. Additionally, MPS1 phosphorylates Thr716 in the MAD1 C-terminal RWD domain to facilitate binding to the N-terminal region of CDC20[16,32,36]. How MPS1 is recruited to kinetochores, however, remains unclear. Previous studies suggested that MPS1 binds to the Ndc80C competitively with microtubules, but contrasting evidence is also available[43,44,62,63]. Recent work in *S. cerevisiae* identified an MPS1 binding site in a region of Ndc80C also implicated in the interaction with the Dam1 complex[64–66]. Whether this site is functionally conserved in humans, however, is not yet clear. The implications of this are further discussed below. As we have not yet been able to reconstitute a robust MPS1-kinetochore interaction in vitro (SG and AM, unpublished results), for our in vitro experiments we either 1) pre-phosphorylated catalysts and kinetochores with MPS1, or 2) added MPS1 directly to an MCC assembly assay.

A second challenge concerns the recruitment of the MAD1:C-MAD2 "template" complex. MAD1:C-MAD2 recruitment requires interactions with BUB1 (facilitated by MPS1 phosphorylation) and with the building block of the kinetochore corona, the ROD-Zwilch-ZW10 (RZZ) complex, and is further regulated by Cyclin B[31,34–36,52,55,67–70]. Yet, we have so far been unable to identify a minimal set of interactions leading to robust MAD1:C-MAD2 recruitment to reconstituted kinetochores in vitro (SS, VC, and AM, unpublished observations; see discussion below), likely because the kinetochore corona is still missing. Kinetochore localization of MAD1:C-MAD2 signals checkpoint activation and mitotic arrest[71–73]. Ectopic retention of MAD1:C-MAD2 on biorientated kinetochores through a fusion with an outer kinetochore subunit is sufficient to cause metaphase arrest[37,74–76].

To target recombinant MAD1:C-MAD2 to reconstituted kinetochores, we engineered a strategy based on inducible dimerization[75,77]. We fused the FKBP-Rapamycin Binding (FRB) domain to the NUF2 subunit of the Ndc80C (Ndc80C^FRB), and the 12-kDa FK506 binding protein (FKBP12, or simply FKBP) to the N-terminus of a MAD1 construct starting at residue 330 and extending to the C-terminal residue 718 (^FKBPMAD1^330-C) (Fig. 2A). This segment of MAD1 has been previously shown to be sufficient for SAC signaling in vitro[16]. We then induced the interaction between Ndc80C^FRB and ^FKBPMAD1^330-C through addition of the small-molecular dimerizer Rapamycin[77]. Analytical size-exclusion chromatography (SEC) confirmed that Ndc80C^FRB and ^FKBPMAD1^330-C:C-MAD2 formed a stoichiometric complex in the presence of Rapamycin (Fig. 2B).

To assess that the recombinant Ndc80C^FRB and ^FKBPMAD1^330-C:C-MAD2 constructs are functional, we electroporated them in HeLa cells previously depleted of endogenous Ndc80C and assessed the kinetochore localization of Ndc80C^FRB and ^FKBPMAD1^330-C, and the duration of mitosis in their presence. In cells depleted of endogenous Ndc80C, electroporation of Ndc80C^FRB in isolation or bound to ^FKBPMAD1^330-C through Rapamycin led to substantial restoration of Ndc80C^FRB kinetochore levels (Fig. 2C, D and Supplementary Fig. 1A). As the endogenous MAD1:C-MAD2 complex is removed from kinetochores at metaphase (Fig. 2E, F, conditions 1-2), we asked if we could observe ^FKBPMAD1^330-C at kinetochores in presence of Rapamycin in cells treated with MG132 to prolong metaphase. Depletion of Ndc80C prevented MAD1 localization to kinetochores (Fig. 2E, F, conditions 3-4), in line with previous observations[51]. Electroporation of Ndc80C^FRB restored localization of endogenous MAD1 in Nocodazole-treated cells, but not at metaphase, as expected (Fig. 2E, F, conditions 5-6). An identical pattern was observed when Ndc80C^FRB and ^FKBPMAD1^330-C:C-MAD2 were co-electroporated but Rapamycin omitted (Fig. 2E, F, conditions 7-8). Conversely, we observed substantial localization of ^FKBPMAD1^330-C at metaphase kinetochores in presence of Rapamycin (Fig. 2E, F, conditions 9-10), indicating that Rapamycin promotes kinetochore localization of ^FKBPMAD1^330-C to Ndc80C^FRB (note that the epitope recognized by the anti-MAD1 is present on both endogenous MAD1 and on ^FKBPMAD1^330-C). Addition of Rapamycin also delayed mitotic exit (Fig. 2G). Albeit less extensive, a mitotic delay was also observed in presence of Ndc80C^FRB or of Ndc80C^FRB and ^FKBPMAD1^330-C in absence of Rapamycin, possibly because Ndc80C^FRB is not fully functional, or because its reduced levels relative to endogenous Ndc80C subtly perturb chromosome alignment, despite evidence that Ndc80C^FRB supports formation of a metaphase plate.

### Engineering an inducible functional SAC scaffold: KNL1

A third challenge concerns the recruitment of BUB1:BUB3. The BUB1:BUB3 complex binds through BUB3, a phospho-aminoacid adapter, to so called MELT motifs on KNL1 after their phosphorylation by MPS1 (Fig. 3A)[47–50,78]. Our kinetochore reconstitutions so far included the C-terminal region of KNL1 (KNL1^C), which binds the second Knl1C subunit ZWINT and mediates kinetochore targeting of the 2316-residue (isoform 2) KNL1 protein[79–83]. This region, however, lacks the MELT repeats (and KI motifs, see below), and is therefore unable to recruit BUB1:BUB3. To generate a KNL1 construct capable of recruiting BUB1:BUB3, we ligated two separately purified fragments of KNL1 encompassing the N- and C-terminal regions. One fragment, KNL1^M5, embeds MELT1, the neighboring KI1 and KI2 motifs, and MELT repeats 12-15, which have been shown to be sufficient to sustain SAC signaling in cells[84]. The other segment, KNL1^C, encompasses the kinetochore-targeting domain (Fig. 3A). We generated a mini-KNL1 protein (KNL1^Bonsai) by covalent ligation of KNL1^M5 and KNL1^C using a SpyTag/SpyCatcher pair[85]. When electroporated into HeLa cells[86], a GFP-labeled version of KNL1^Bonsai (^EGFPKNL1^Bonsai) decorated kinetochores in mitotic cells, indicating that our recombinant construct retains kinetochore localization properties similar to those of endogenous KNL1 (Supplementary Fig. 1B–D). Seemingly limited stability of ^EGFPKNL1^Bonsai in absence of endogenous KNL1 prevented us from carrying out the same experiment in absence of endogenous KNL1.

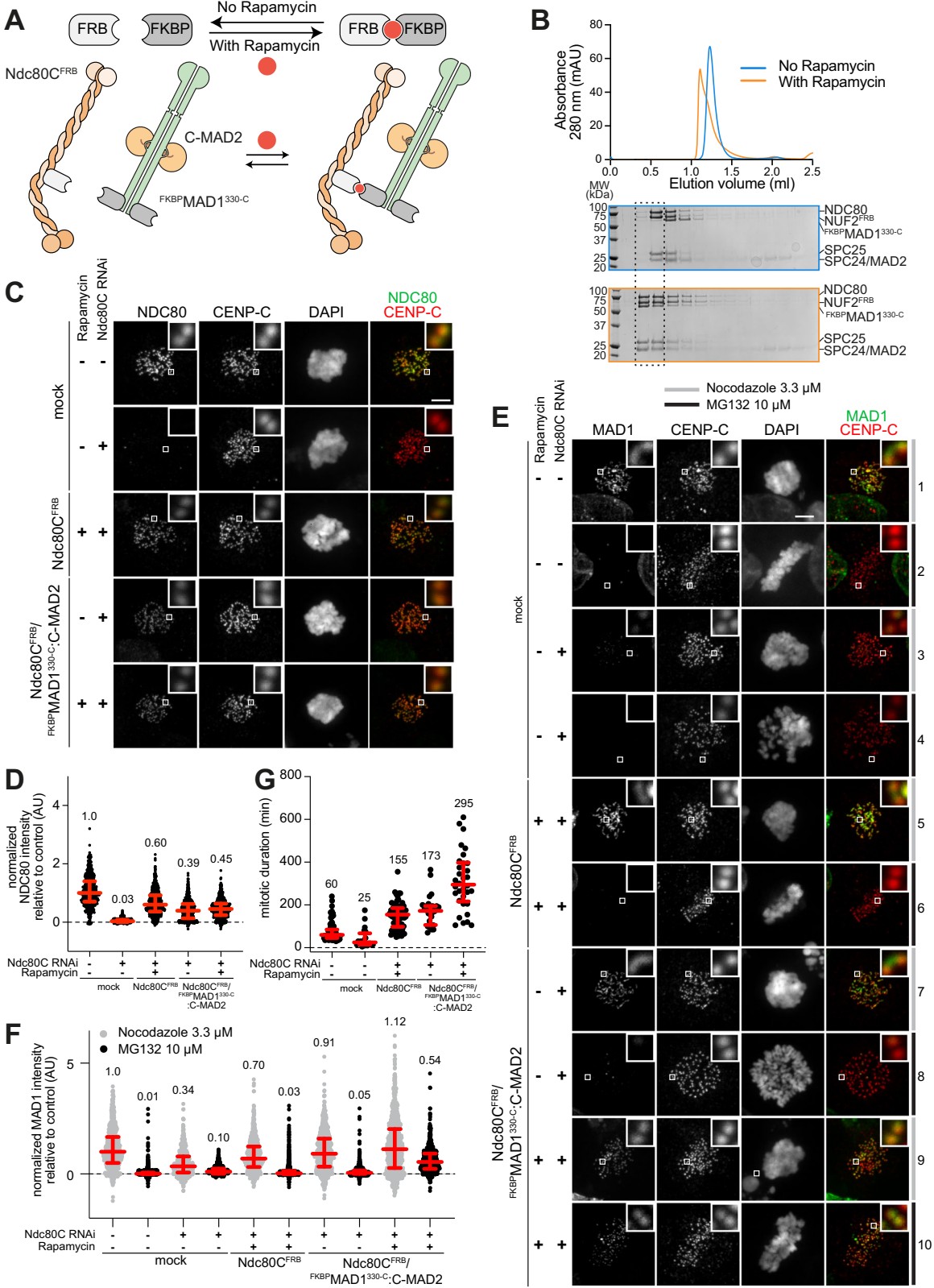

The N-terminal region of CENP-C provides a point of attachment for the KMN network to the kinetochore[87,88]. To develop a binding assay to monitor the interaction of various moieties with the outer kinetochore, we immobilized a GST fusion of CENP-C[1–71]. We then monitored retention of outer kinetochore proteins on the solid phase (as illustrated schematically in Fig. 3C). $^{GST}$CENP-C[1–71] bound the entire recombinant KMN network but failed to retain the KNL1[M5] fragment.

Conversely, both the KNL1[C] and the KNL1[Bonsai] constructs were effectively retained on solid phase (Fig. 3D). Our reconstituted kinetochore incorporating KNL1[Bonsai] (herewith indicated as $^{R}$KT, for "reconstituted kinetochore") may also be expected to recruit BUB1:BUB3 upon phosphorylation of the five MELT repeats with MPS1. Below, we also demonstrate this prediction. We also asked if $^{R}$KT, after immobilization on solid phase, recruited $^{FKBP}$MAD1[330-C]:C-MAD2 in the presence of

**Fig. 2 | Engineering MAD1 recruitment. A** Scheme of FRB and FKBP dimerization through Rapamycin (red circle). With Rapamycin $^{FKBP}$MAD1$^{330-C}$:C-MAD2 binds FRB fused C-terminally to NUF2 in Ndc80C$^{FRB}$. **B** Elution profiles and SDS PAGE of fractions of SEC experiments on Superose 6 5/150 column with stoichiometric $^{FKBP}$MAD1$^{330-7I8}$:C-MAD2 and Ndc80C$^{FRB}$ (3 μM) with/without 10 μM Rapamycin. Representative elution profile from three repeats. **C** Immunofluorescence of HeLa cells treated with Rapamycin 500 nM or Ndc80C RNAi (see "Methods") and electroporated as indicated (Ndc80C$^{FRB}$ or Ndc80C$^{FRB}$/$^{FKBP}$MAD1$^{330-C}$:C-MAD2). CENP-C is a kinetochore resident. Scale bar 5 μm. **D** Kinetochore levels of endogenous Ndc80C or Ndc80C$^{FRB}$ normalized to endogenous levels (no RNAi, no Rapamycin). Number of kinetochores from two independent experiments: $n = 553$ for control, $n = 709$ for Ndc80C RNAi, $n = 642$ for Ndc80C$^{FRB}$, $n = 565$ for Ndc80C$^{FRB}$/$^{FKBP}$MAD1$^{330-C}$:C-MAD2 no Rapamycin, $n = 545$ for Ndc80C$^{FRB}$/$^{FKBP}$MAD1$^{330-C}$:MAD2 with Rapamycin. Here and in panel F, red bars represent median, indicated above the plots, and interquartile range of normalized single kinetochore intensities. **E** Immunofluorescence of HeLa cells treated with Rapamycin or Ndc80C RNAi and

electroporated as indicated (Ndc80C$^{FRB}$ or Ndc80C$^{FRB}$/$^{FKBP}$MAD1$^{330-C}$:C-MAD2). Localization of MAD1 was compared in SAC-on cells (Nocodazole, conditions 1, 3, 5, 7, and 9) or SAC-off cells (MG132, conditions 2, 4, 6, 8, and 10). Scale bar 5 μm. **F** Quantification of MAD1 normalized to endogenous levels. Number of kinetochores from two independent experiments: $n = 728$ for condition 1, $n = 1487$ for 2, $n = 630$ for 3, $n = 871$ for 4, $n = 658$ for 5, $n = 1069$ for 6, $n = 758$ for 7, $n = 768$ for 8, $n = 898$ for 9, $n = 714$ for 10. **G** Duration of mitosis in HeLa cells treated with Rapamycin or Ndc80C RNAi as described (see Methods) and electroporated as indicated (Ndc80C$^{FRB}$ or Ndc80$^{FRB}$/$^{FKBP}$MAD1$^{330-C}$:C-MAD2). Cells arrested in G2 phase were released into mitosis and imaged for 18 h. Mitotic duration was measured using cell morphology and DNA condensation in two independent experiments. Red bars represent median with interquartile range. Cells analyzed in two independent experiments: $n = 113$ for control, $n = 21$ for Ndc80C RNAi, $n = 74$ for Ndc80C$^{FRB}$, $n = 29$ for Ndc80C$^{FRB}$/$^{FKBP}$MAD1$^{330-C}$:C-MAD2 no Rapamycin, $n = 35$ for Ndc80C$^{FRB}$/$^{FKBP}$MAD1$^{330-C}$:C-MAD2 with Rapamycin.

Ndc80C$^{FRB}$ and Rapamycin. Indeed, $^{FKBP}$MAD1$^{330-C}$:C-MAD2 was retained on solid phase upon addition of Rapamycin (Fig. 3E), indicating that we have obtained an inducible system for recruiting SAC signaling components on reconstituted kinetochores. Under the same conditions, we failed to observe binding of MAD1:C-MAD2 (SS and AM, unpublished observations), indicating that its robust recruitment necessitates of additional determinants, probably in the kinetochore corona.

## Catalytic activation of SAC signaling by reconstituted kinetochores

Next, we tested if adding the $^R$KT to the SAC catalysts and the SAC substrates affected the rate of MCC production. For these experiments, we used our previously described FRET sensor consisting of $^{CFP}$BUBR1 as fluorescence donor and O-MAD2$^{TAMRA}$ as fluorescence acceptor[16]. In this system, incorporation of MAD2 and BUBR1 in MCC is contingent on the presence of CDC20, so that the increase in FRET signal over time detects the time-dependent accumulation of MCC (Fig. 4A).

Previously, we demonstrated that addition of MAD1:C-MAD2 and BUB1:BUB3 (each at 100 nM) pre-phosphorylated with MPS1 kinase (50 nM) greatly accelerates MCC assembly in the presence of CDC20 (500 nM)[16,23]. Indeed, very rapid accumulation of MCC was observed under these conditions (Fig. 4B, red curve. Supplementary Table 1 reports concentrations of reagents used in MCC assembly assays throughout this work). Addition of $^R$KT (20 nM) that had also been pre-treated with MPS1 (pre-phosphorylated $^R$KT, indicated as $^R$KT-(P)) to phosphorylate the five MELT motifs on KNL1$^{Bonsai}$ further increased the rate of MCC production (Fig. 4B, blue curve). The acceleration of the rate of MCC production by $^R$KT-(P)) was not complementary to that provided by catalysts, as it was completely dependent on the presence of catalysts (Supplementary Fig. 1E, F). Thus, at an identical concentration of MCC substrates and pre-phosphorylated catalysts, addition of $^R$KT-(P) further accelerates MCC production, seemingly recapitulating in vitro a role of kinetochores in SAC signaling.

Under the conditions that support rapid accumulation of MCC even in the absence of kinetochores in our FRET assay ([16,23] and Fig. 4B), the concentrations of catalysts are higher than those estimated for human cells. MAD1, for instance, may be present at 10-20 nM in cells[19,89] but was used at 100 nM in Fig. 4B. To assess how the rate of MCC accumulation scales with catalyst concentration, we pre-phosphorylated the catalysts with MPS1 and diluted them in the MCC FRET assays. MCC accumulation progressed very rapidly at the highest concentrations of catalysts (40 nM or 100 nM for both $^{FKBP}$MAD1$^{330-C}$:C-MAD2 and BUB1:BUB3), and the rate was apparently insensitive to the lower CDC20 concentration (from 500 nM to 100 nM, compare experiments in Fig. 4B, C), i.e. a concentration equimolar with the concentration of MAD2 and BUBR1:BUB3 and closer to the cellular concentration of CDC20, estimated between 100 and 285 nM[10,11]. On

the other hand, the rate of MCC assembly decreased considerably at lower concentrations of pre-phosphorylated catalysts (Fig. 4C).

At 20 nM $^{FKBP}$MAD1$^{330-C}$:C-MAD2 and 40 nM BUB1:BUB3, an intermediate rate of stimulation of MCC assembly was observed (Fig. 4C). We settled on this concentration of catalysts to test the role of pre-phosphorylation and kinetochores in MCC assembly. We reasoned that pre-phosphorylation may bypass a requirement for kinetochore-localized kinase activity for effective accumulation of phosphorylated scaffolds and catalysts. To test this, we monitored the rate of MCC production in the absence of MPS1 pre-phosphorylation of catalysts. In the absence of pre-phosphorylation, a concentration of catalysts of 20 nM $^{FKBP}$MAD1$^{330-C}$:MAD2 and 40 nM BUB1:BUB3 failed to promote rapid accumulation of MCC assembly, probably because MPS1-mediated phosphorylation of these reagents at these concentration proceeds too slowly (Fig. 4D, gray curve). Addition of 20 nM $^R$KT strongly stimulated MCC assembly (Fig. 4D, blue curve). Stimulation was dependent on addition of Rapamycin, indicating that it requires MAD1:C-MAD2 accumulation on kinetochores (green curve in Fig. 4D and Supplementary Fig. 2A, B). It was also dependent on ATP, required for kinase activity (Supplementary Fig. 1G). Stimulation also required intact KNL1$^{Bonsai}$, because $^R$KTs incorporating unfused KNL1$^C$ and KNL1$^{M5}$ (split KNL1$^{Bonsai}$, Fig. 4E) did not support accelerated MCC production (Fig. 4F; green curve). Finally, K$^C$MN was unable to stimulate MCC assembly, indicating that functional MELT repeats on the K$^{M5}$ fragment are required (Fig. 4F, purple curve).

Collectively, these observations are consistent with the idea that incorporation in the $^R$KT of MAD1:C-MAD2 (through Rapamycin-induced binding to Ndc80C) and BUB1:BUB3 (through MPS1-mediated phosphorylation of MELT repeats) greatly enhances the rate of MCC assembly. Confirming the importance of MELT phosphorylation by MPS1, adding $^R$KT-(P) to catalysts at the standard concentration resulted in much more rapid MCC production in comparison to when $^R$KT was not pre-phosphorylated (Supplementary Fig. 1H). Even if not pre-phosphorylated, however, the $^R$KT accelerated MCC assembly relative to a condition in which it was omitted altogether (Supplementary Fig. 1H). Omission of the $^R$KT in the absence of pre-phosphorylation of the catalysts did not allow rapid assembly of MCC (Supplementary Fig. 1H), even at very high concentrations of catalysts (Supplementary Fig. 2C–G). We added MPS1 to $^R$KT and catalysts, and monitored phosphorylation of the MELT repeats, of BUB1, and MAD1 (Supplementary Fig. 1I). Probably because our $^R$KT do not recruit MPS1, phosphorylation of these substrates reached completion only after ~60 minutes (Supplementary Fig. 1I), likely explaining why pre-incubation is effective, and implying that only a fraction of $^R$KT and catalysts contributes to accelerated MCC assembly under the conditions of the assay.

The constitutive centromere-associated network (CCAN) is a group of 16 proteins in humans that connects the outer kinetochore

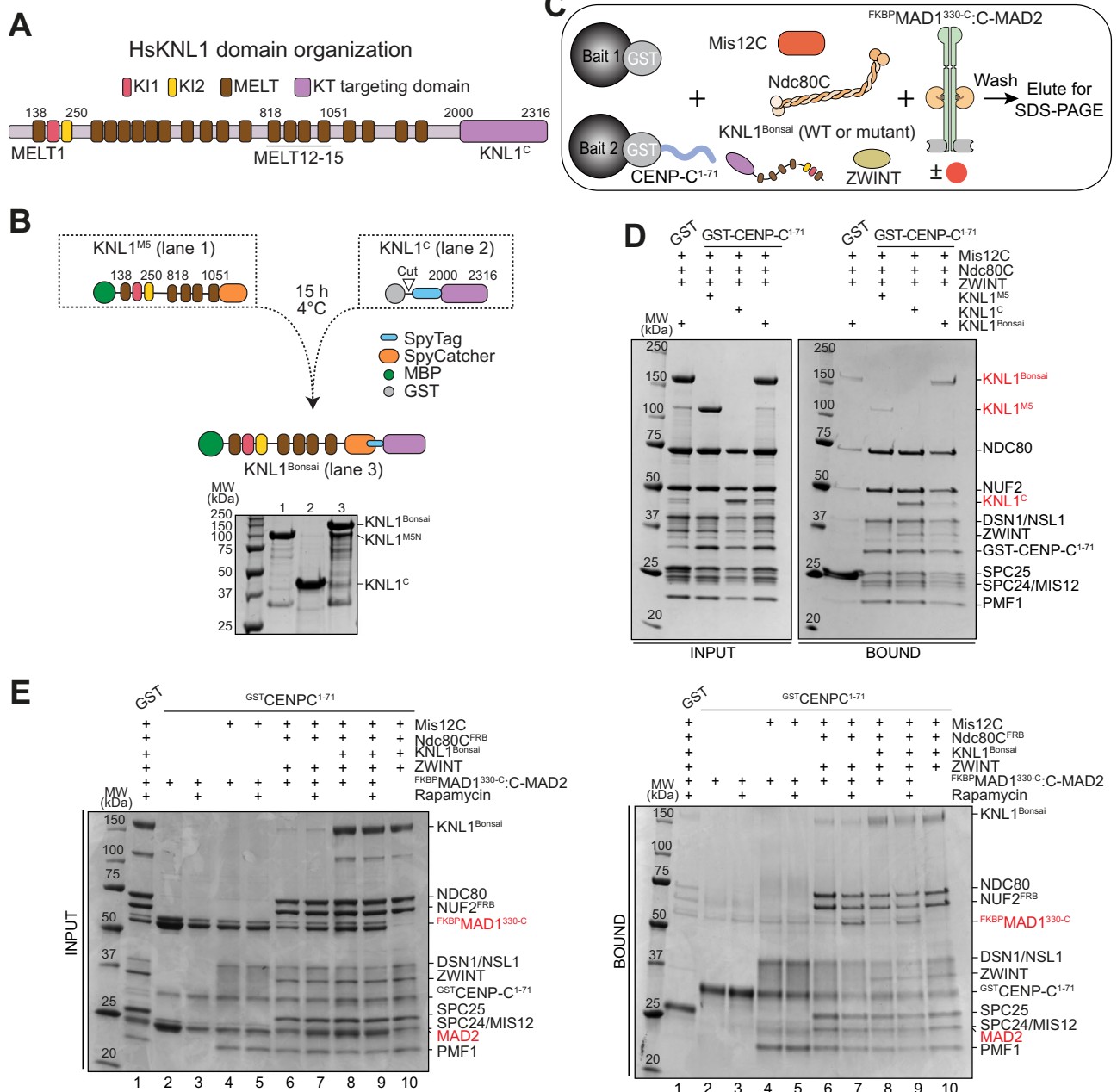

**Fig. 3 | Engineering KNL1. A** Scheme of motifs and domains of human KNL1. Numbering is according to isoform 2 of KNL1 (residues 1-2316, with residues 84-109 missing relative to the more canonical isoform 1 of 2342 residues). We opted to use isoform 2 numbering to facilitate a comparison with previously used constructs[99]. KI1 and KI2, lysine (K) – isoleucine (I) motifs; MELT, methionine (M) – glutamic acid (E) – leucine (L) – threonine (T) motifs. The KNL1$^C$ fragment contains tandem RWD domains (RING finger-containing proteins, WD-repeat-containing proteins and yeast DEAD (DEXD)-like helicases)[79]. **B** Strategy for generating KNL1$^{Bonsai}$ through fusion of fragments fused to SpyTag or SpyCatcher. The KNL1$^{M5}$ fragment is an engineered fusion of residues 138-250 and 818-1051. The fusion reaction was performed at least three times with essentially identical results. **C** Scheme of solid phase binding assay monitoring interactions of outer kinetochore and SAC proteins. **D** KNL1$^C$ and KNL1$^{Bonsai}$ bind Mis12C immobilized on solid phase, whereas only traces of KNL1$^{M5}$ were detected, as expected. This experiment and the experiment in panel E are representative of three repeats. **E** SDS PAGE result of pulldown with GST-CENP-C$^{1-71}$ and $^R$KT with NUF2$^{FRB}$ (3 μM) testing for binding of $^{FKBP}$MAD1:C-MAD2 (6 μM) with or without Rapamycin.

KMN network to the centromere[90]. We modified our solid phase binding assay by replacing $^{GST}$CENP-C$^{1-71}$ with $^{GST}$CENP-C$^{1-544}$ (Supplementary Fig. 3A). In addition to binding Mis12C, this segment of CENP-C also binds the CCAN complex[91–97]. As expected, both the $^R$KT and CENP11, a CCAN sub-assembly consisting of 11 subunits (the 5-subunit CENP-OPQUR subcomplex, the 4-subunit CENP-HIKM complex, the 2-subunit CENP-LN complex) previously shown to be sufficient to bind CENP-C (another CCAN subunit binding to

CENP11, thus giving rise to CENP12)[98], were retained on solid phase in the presence of $^{GST}$CENP-C$^{1-544}$ (Supplementary Fig. 3B). To assess a possible role of CCAN in MCC assembly, we ran the MCC assembly assay in the presence of catalysts, $^R$KT, and Rapamycin and asked whether addition of CENP12 enhanced the MCC assembly rate. There was no difference in the rate of assembly of MCC with or without CENP12, suggesting that the latter does not stimulate MCC assembly (Supplementary Fig. 3C).

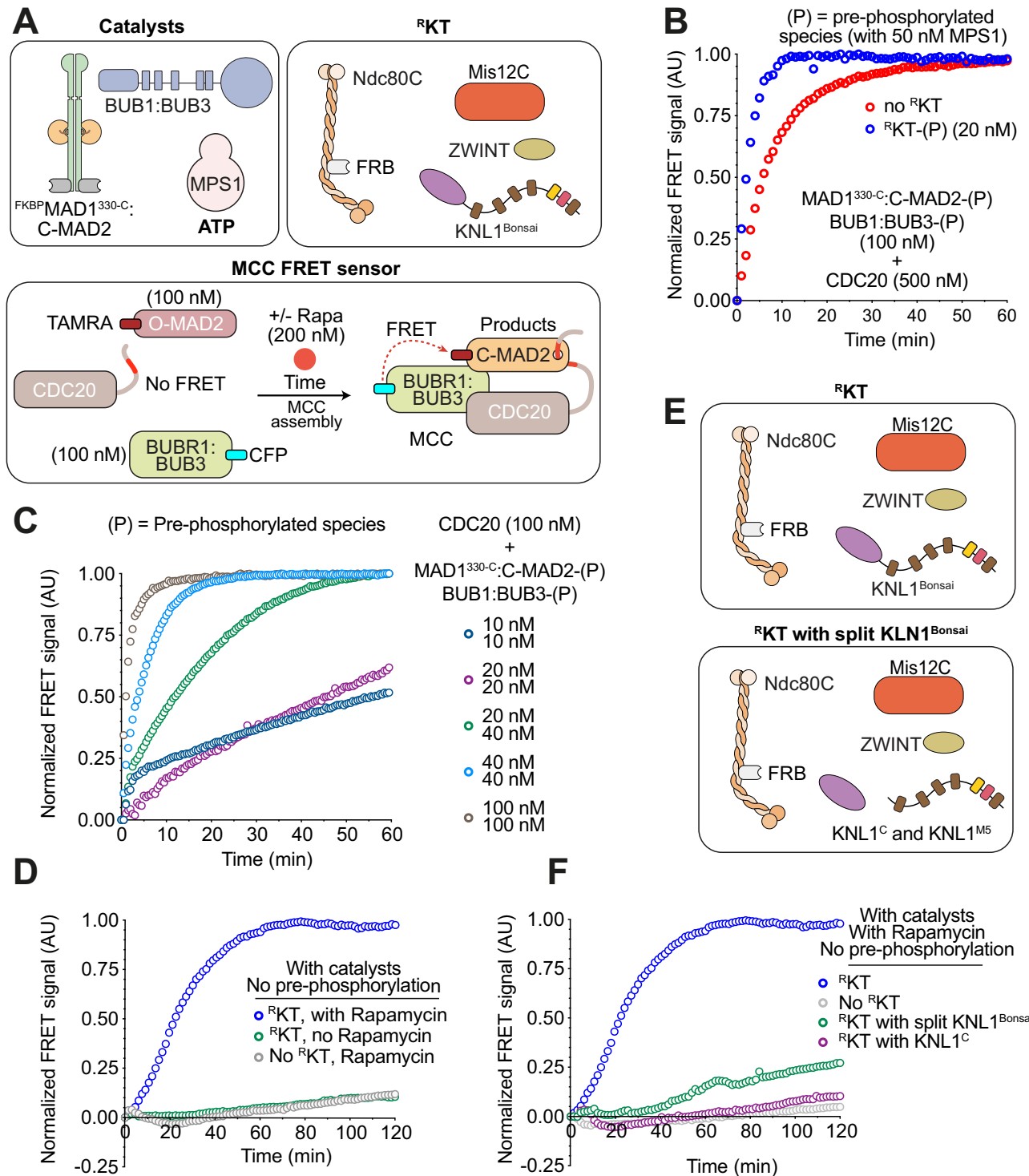

## Relative potency of MELT and KI motifs of KNL1

The MELT repeat unit is flanked by additional motifs, including the Thr-X-X-Ω (X = any aminoacid, Ω = Phe or Tyr) and Ser-His-Thr (SHT) motifs, positioned N- and C-terminally to the MELT sequence, respectively. The SHT motif contributes to recruiting BUB1:BUB3 to the kinetochore[84]. Two additional motifs, KI1 and KI2, interact respectively with the tetratricopeptide repeat (TPR) domains of BUB1 and BUBR1. Together with the first MELT motif (MELT1) of KNL1, the KI1 and KI2 motifs form a reinforced unit for BUB1:BUB3 recruitment and SAC signaling[78,81,99–101]. We carried out additional experiments to assess the role of MELT phosphorylation in the catalytic assembly of MCC. First, we generated a set of KNL1 variants and mutants as

indicated in Fig. 5A, mixed them with Mis12C and Ndc80C, and immobilized them on solid phase through [GST]CENP-C[1–71]. Then, we tested the ability of the various KNL1 species to bind and retain BUB1:BUB3 on solid phase upon phosphorylation with MPS1 kinase. KNL1[C] did not bind BUB1:BUB3 above background levels, as expected for a construct lacking MELT motifs altogether (Fig. 5B, lane 2; the negative control is in lane 1). Conversely, KNL1[Bonsai] or its MPS1 pre-phosphorylated form bound BUB1:BUB3, with pre-phosphorylation leading to apparently superstoichiometric binding (lanes 3-4). Without MPS1 pre-phosphorylation, mutation of the KI1 and KI2 motifs (K[KI1+2]MN) led to a strong decrease of BUB1:BUB3 binding (lane 5), consistent with the idea that KI1 and KI2 provide a high-affinity,

**Fig. 4 | KMN supports catalytic MCC assembly. A** Scheme of catalysts, [R]KT, and FRET sensor monitoring MCC assembly[16,23]. [CFP]BUBR1 and O-MAD2[TAMRA] are donor and acceptor, respectively. CFP and TAMRA are closely positioned in MCC, allowing FRET (quantified as sensitized TAMRA fluorescence emission). **B** Sensitized fluorescence emission (FRET) normalized to curve's maximum for MCC sensor in presence of indicated concentration of MPS1-pre-phosphorylated catalysts, and supplemented with (blue) or without (red) pre-phosphorylated [R]KTs (20 nM) in presence of 200 nM Rapamycin. ATP (2 mM) and MgCl$_2$ (10 mM) were added before starting the experiment. All catalytic components were pre-phosphorylated with MPS1. In all FRET experiments, the Y-axis represents fluorescence emission intensity of FRET acceptor, indicated as "Normalized FRET signal (AU)". AU is arbitrary units, and we show it to emphasize the original signal was in arbitrary units. The blue and red curve were normalized to their own maximum. **C** Sensitized fluorescence emission (FRET) normalized to curve's maximum for MCC sensor in presence of indicated concentrations of MPS1-pre-phosphorylated catalysts. [R]KTs

were omitted. Dark blue and purple curves did not plateau during observation and their values were normalized to the maximum value of the light brown-color curve. **D** Sensitized fluorescence emission (FRET) of MCC sensor in the presence of unphosphorylated catalysts (50 nM MPS1; 20 nM [FKBP]MAD1[330-C]:C-MAD2; 40 nM BUB1:BUB3) and unphosphorylated [R]KT (20 nM). The blue curve (also shown in **F** and in Supplementary Fig. 1G) was normalized to its own maximum. Fluorescence values of the other curves were normalized to the maximum value of the blue curve. The gray curve is also shown in Supplementary Fig. 1G. **E** Variants of the [R]KT used for controls in **F**. **F** Sensitized fluorescence emission (FRET) of MCC sensor in the presence of unphosphorylated catalysts (same concentration as in **D**, unphosphorylated [R]KT (same curve as in panel **D**), and the [R]KT variants presented in **E**. Fluorescence values were normalized to maximum of the blue curve. Panels reporting time-dependent changes in FRET signal are from single measurements and representative of at least three independent technical replicates of the experiment.

phosphorylation-independent binding site for BUB1:BUB3[81,99,100]. MPS1 pre-phosphorylation of K[KII+2]MN enhanced BUB1:BUB3 binding, likely through phosphorylation of the MELT motifs (lane 6). Finally, a mutant in which both the MELT and the adjacent SHT motifs were mutated (to MELA and AHA, respectively; K[M5-A]MN) was also a weak BUB1:BUB3 binder, and as expected pre-phosphorylation did not rescue this effect of the mutations, confirming that MPS1 phosphorylation promotes BUB1:BUB3 binding through the MELT repeats (lanes 7-8). Collectively, these binding experiments indicate that [R]KTs containing KNL1[Bonsai] and its variants behave as expected based on previous binding analyzes[36,47–49,78,81,84,99,102–104].

Next, we assessed how different variants of KNL1[Bonsai] in [R]KTs affected the rate of MCC accumulation in our FRET assay. Both the KNL1[M5/A] and the KNL1[KII+2/A] mutants caused a strong reduction of the rate of MCC accumulation (Fig. 5C; the residual activation was dependent on Rapamycin, as shown in Supplementary Fig. 2A, B). These results imply that the KI1 and KI2 motifs contribute substantially to catalytic MCC assembly in the context of KNL1[Bonsai] and that they play a role in MCC assembly even in the presence of five MELT motifs in this construct. The results are consistent with the idea that the combination of closely spaced KI1 and MELT1 creates a composite binding site that binds BUB1:BUB3 cooperatively and with high affinity, decreasing its turnover and providing a stable platform for the SAC catalytic apparatus, as previously suggested by an in vivo analysis[99]. In agreement with this idea, a construct containing only the first MELT repeat and the two adjacent KI motifs (KNL1[Bonsai-M1], generated as indicated schematically in Supplementary Fig. 4A), bound BUB1 on solid phase (Supplementary Fig. 4B, C) and was a robust activator of MCC assembly in the presence of Rapamycin but not in its absence (Fig. 5D and Supplementary Fig. 2A-B).

#### Role of BUB1 in MCC assembly

BUB1 contains sequence motifs that promote interactions with the kinetochore and with other SAC components (Supplementary Fig. 4D), and that contribute to catalytic MCC assembly in the absence of reconstituted kinetochores[23]. We were curious to verify if these motifs become dispensable in the presence of kinetochores. Interaction of two single-helices promotes the interaction of the BUB1 and BUBR1 paralogs and kinetochore recruitment of BUBR1[105]. In our [GST]CENP-C[1–71] pulldown assay, a mutant lacking the helix (BUB1[Δhelix]) failed to recruit BUBR1 (Fig. 6A, B). Replacing BUB1 with BUB1[Δhelix] did not cause any effect on the rate of MCC formation in the presence of [R]KT in vitro (Fig. 6C, D). This result recapitulates our previous results in vivo[105,106] and demonstrates that the recruitment of BUBR1:BUB3 to the catalytic platform is not required for rapid MCC assembly.

Using its ABBA, KEN, and CM1 motifs, BUB1 positions CDC20 in close proximity to the MAD1 coiled-coil[23,25]. In our pulldown assay, a BUB1[KEN-ABBA] mutant (where both the KEN and ABBA motifs were mutated to alanine, see "Methods") retained lower amounts of CDC20

relative to wild type BUB1. In the MCC assembly assay, the same mutant demonstrated a very strong decrease in the rate of MCC assembly (Fig. 6C, E). Further dissection of this BUB1 region demonstrated that mutation of KEN2 had the strongest deleterious effect on the rate of MCC assembly, followed by mutation of KEN1 and mutation of the ABBA motif (Supplementary Fig. 4E–G).

The CM1, a target of CDK1:Cyclin B and MPS1 phosphorylation, contributes to MAD1:MAD2 recruitment to the kinetochore[30–32,34–36,68]. In the MCC assembly assay, deletion of CM1 led to a very significant reduction in the rate of MCC assembly, but did not abrogate catalytic MCC assembly (Fig. 6C, F). In vivo, inhibition of the BUB1:MAD1 binding interaction is incompatible with robust SAC signaling[30,34,37,54,75]. Our observation that MCC assembly in vitro is only partially impaired when MAD1 and BUB1 cannot interact suggests that kinetochores, by recruiting BUB1 and MAD1:MAD2 in close proximity, may partly compensate for the loss of an interaction of the catalysts. Nonetheless, the discrepancy also suggests that our system may not yet fully recapitulate all properties of SAC catalysis.

#### BUB1 and MAD1:C-MAD2 are an integrated platform for catalytic MCC assembly by MPS1

As discussed earlier, phosphorylation of KNL1 by MPS1 is required for a robust rate of MCC assembly. MPS1 has also been shown to phosphorylate Thr461 in the CM1 motif of BUB1, which in turn promotes the interaction with MAD1 and its kinetochore recruitment[30,31,34,36]. In addition, MPS1 phosphorylates residue Thr716 at the C-terminus of MAD1, an event that activates the catalytic function of MAD1[16,23,33,36,52]. To investigate the specific role of MPS1 kinase activity in our MCC assembly assay, we conducted a comprehensive analysis to dissect individual phosphorylation requirements for this kinase. Addition of Reversine, a potent inhibitor of MPS1 kinase activity[107], ablated catalytic MCC assembly (Fig. 7A). At high concentrations of pre-phosphorylated catalysts and in the absence of kinetochores, individual alanine mutations of Ser459 and Thr461 on BUB1 did not inhibit MCC catalysis in vitro[23]. We combined mutations of Ser459 and Thr461 in a single mutant and performed the MCC assembly reaction in the presence of kinetochores. We observed a moderate decline in the rate of MCC assembly (Fig. 7B, C). Additionally, when we replaced wild-type MAD1 with the MAD1[T716A] mutant in the MCC assembly assay, we also observed a moderate decrease of the rate of MCC assembly (Fig. 7B, C). When the phospho-alanine mutants MAD1[T716A] and BUB1[S459A-T461A] were combined, the rate of MCC assembly dropped substantially, becoming similar to the rate in the absence of MPS1 kinase (Fig. 7B, C).

#### An overview of MCC assembly

In this study, we have pushed the envelope for in vitro biochemical reconstitutions by combining two major reconstitutions in a single test tube, a reconstitution of the kinetochore and a reconstitution of the SAC. This approach allowed us to investigate the effects of the

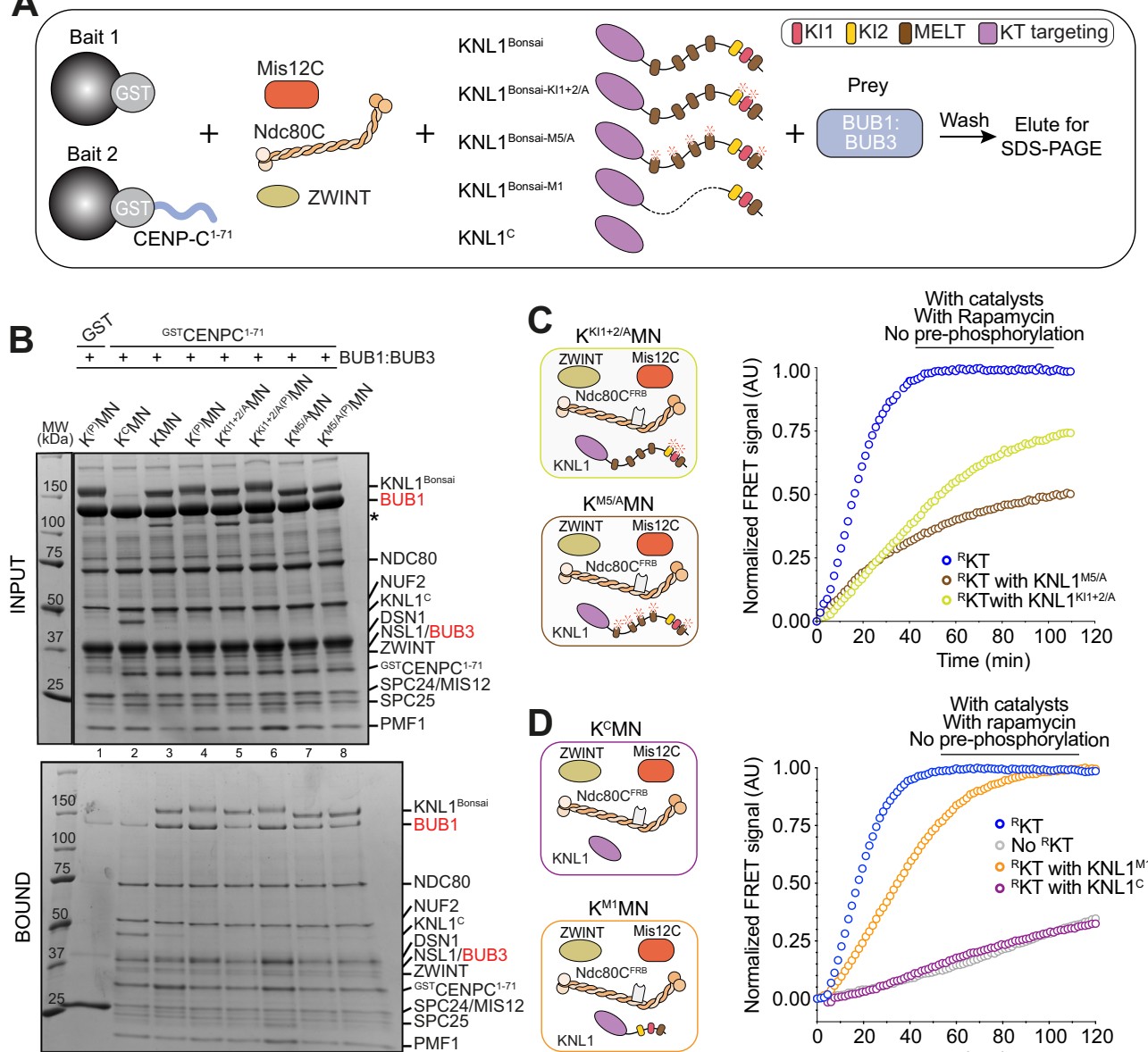

**Fig. 5 | KNL1^Bonsai brings BUB1:BUB3 in proximity of MAD1:MAD2. A** Scheme of GST pulldown assays with ^GSTCENP-C^1–71 and ^RKT (3 μM) assembled with the indicated versions of Knl1C and further incubated with BUB1:BUB3 (10 μM) as pray. **B** SDS PAGE of pulldown experiment schematized in **A** and containing ^GSTCENP-C^1–71 and ^RKT at 3 μM with different KNL1 mutants and BUB1:BUB3 at 10 μM as indicated above each lane. The asterisks indicate excess of KNL1^M5 fragments not fused to KNL1^C. The experiment was performed three times with essentially identical results. **C** Sensitized fluorescence emission (FRET) of MCC sensor in the presence of unphosphorylated catalysts and unphosphorylated ^RKT containing the indicated KNL1 variants, supplemented at the same concentrations used in Fig. 4D. The blue curve was normalized to its maximum. The other curves did not plateau during the experiment and were normalized to the maximum of the blue curve. **D** Sensitized fluorescence emission (FRET) of MCC sensor in the presence of unphosphorylated catalysts and unphosphorylated ^RKT containing the additional indicated KNL1 variants, supplemented at the same concentrations used in Fig. 4D. The blue and orange curves were normalized to their own maximum. The gray and purple curves were normalized to the maximum of the blue curve. Panels reporting time-dependent changes in FRET signal are from single measurements and representative of at least three independent technical replicates of the experiment.

kinetochore on SAC signaling. After solving several challenges to allow robust recruitment of SAC components to ^RKT, we were able to demonstrate a role of ^RKTs in the acceleration of MCC assembly. This acceleration is contingent on the recruitment of MAD1:C-MAD2 and BUB1:BUB3 to ^RKTs, as it requires Rapamycin (to recruit MAD1:C-MAD2) and the MELT and KI repeats of KNL1 (to recruit BUB1:BUB3). Furthermore, acceleration of MCC assembly is contingent on the interaction of MAD1:MAD2 and BUB1:BUB3, as mutations in the CM1 motif of BUB1 affect the rate of MCC assembly. Thus, even within ^RKT,

the juxtaposition of MAD1:C-MAD2 and BUB1:BUB3 contributes substantially to catalysis (Fig. 7D), indicating that kinetochores do not simply work by concentrating different SAC catalysts in a single scaffold.

We resorted to an artificial dimerization system to trigger MAD1:C-MAD2 recruitment to ^RKTs, as we have not yet been able to recapitulate robust recruitment of MAD1:C-MAD2 with unmodified purified components. While the precise reasons for this will require further investigations, we suspect that introduction of a proper

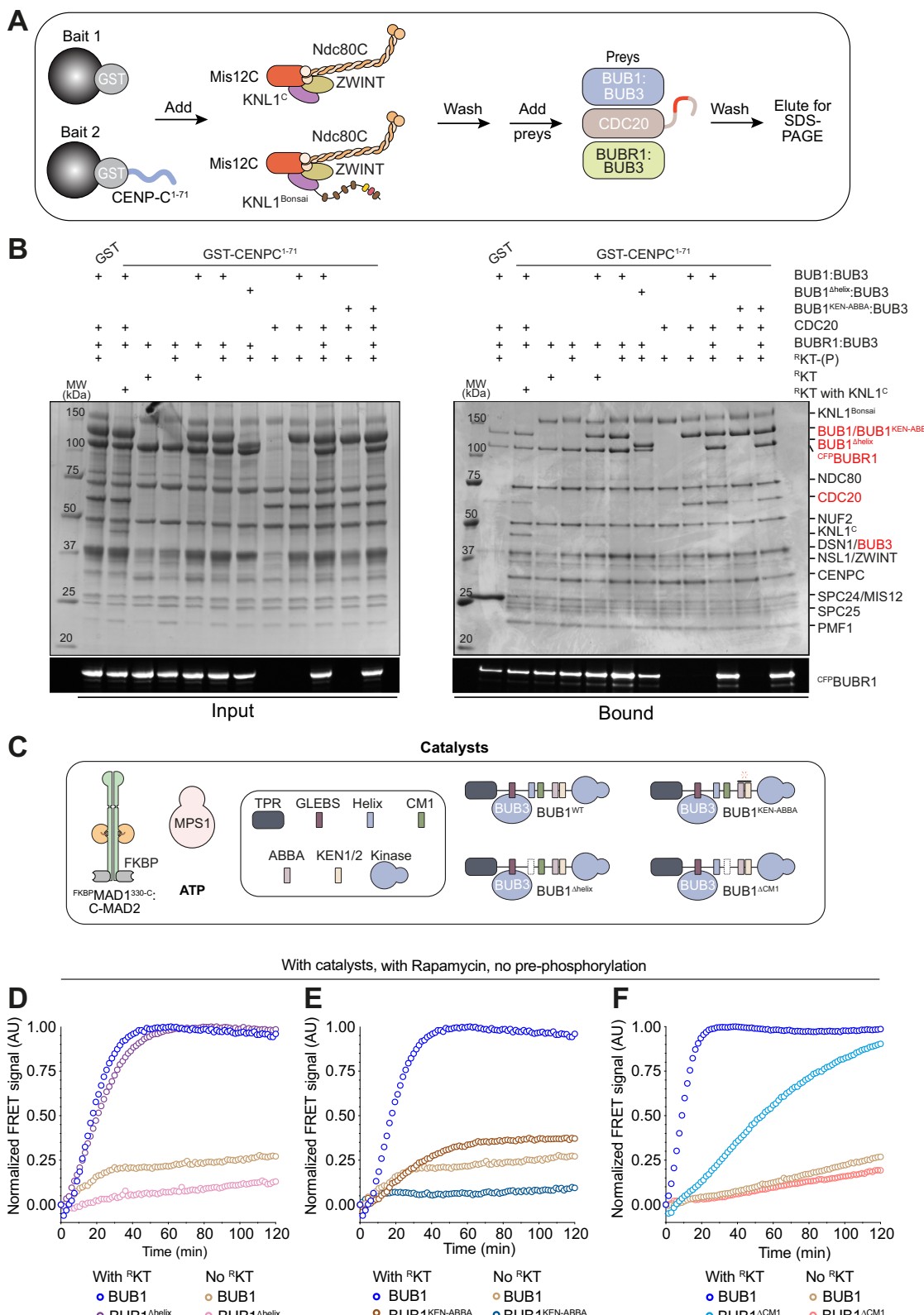

kinetochore corona may be required. We surmise that the achievement of "natural" recruitment conditions in our reconstitutions may promote an even more favorable reciprocal positioning of MAD1:C-MAD2 and BUB1:BUB3. Collectively, our observations are consistent with the idea that efficient SAC signaling requires the proximity and direct interaction of BUB1:BUB3 and MAD1:C-MAD2 in a single catalytic complex[30,37,54,75]. Nonetheless, we cannot exclude that the sole function

of kinetochores is simply to recruit the SAC's catalytic components, without major restrains on their reciprocal positioning.

Crucially, BUB1:BUB3 becomes recruited to the kinetochore directly through an interaction with the MELT repeats of KNL1, triggered by their MPS1-dependent phosphorylation. Only few of the 19 MELT motifs of human KNL1 are occupied by BUB1:BUB3 during SAC activation[78,84,99,102–104]. Engineered KNL1 constructs containing subsets

**Fig. 6 | Influence of BUB1:BUB3 interaction with other SAC components in catalysis. A** Scheme of pulldown assays with [GST]CENP-C[1–71], [R]KT or [R]KT assembled with KNL1[C] (rather than KNL1[Bonsai], each at 3 µM), and prays including BUB1:BUB3 (10 µM), [CFP]BUBR1:BUB3 (8 µM), and CDC20 (6 µM). **B** SDS PAGE of pulldowns with baits and preys indicated above each lane. Pre-phosphorylation of the [R]KTs species by MPS1 (50 nM) is indicated with (P). Coomassie staining is shown above in-gel fluorescence (488 nm) below. All baits were incubated with an at least 3-fold molar excess of BUB1:BUB3, [CFP]BUBR1:BUB3 and CDC20. Efficient phosphorylation of KNL1 and KNL1[KI1+2/AA] was confirmed by a shift in migration in SDS-PAGE of the KNL1 band. No shift of KNL1[M5-A] upon phosphorylation with MPS1 was observed. The experiment was performed three times with essentially identical results. **C** Scheme of the experimental setup of the following assays. **D** MCC FRET assay monitoring the assembly of MCC without pre-phosphorylation of the components by MPS1, in the presence of catalysts and Rapamycin, and with [R]KT (blue curve, also shown in

**E** or without [R]KT (light brown curve, also shown in **E**. Where indicated, the BUB1[Δhelix] mutant replaced wild-type BUB1 and was tested with [R]KT (thistle curve) or without [R]KT (pink curve). **E** MCC FRET assay monitoring the assembly of MCC without pre-phosphorylation of the components by MPS1, in the presence of catalysts and Rapamycin, and with [R]KT (blue curve, also shown in **D** or without [R]KT (light brown curve, also shown in **D**. Where indicated, the BUB1[KEN-ABBA] mutant replaced wild-type BUB1 and was tested with [R]KT (dark brown curve) or without [R]KT (dark blue curve). **F** MCC FRET assay monitoring the assembly of MCC without pre-phosphorylation of the components by MPS1, in the presence of catalysts and Rapamycin, and with [R]KT (blue curve, also shown in Fig. 7A and C) or without [R]KT (light brown curve). Where indicated, the BUB1[ΔCMI] mutant replaced wild-type BUB1 and was tested with [R]KT (light blue curve) or without [R]KT (salmon curve). Panels reporting time-dependent changes in FRET signal are from single measurements and representative of at least three independent technical replicates of the experiment.

of MELT repeats, including the single MELT1 with its neighboring KI1 and KI2 motifs, or an array of six MELTs, were shown to be sufficient for a very robust SAC response[78,84,99,103,104]. Thus, we believe that KNL1[Bonsai], with its five MELT motifs including MELT1 (and associated KI1 and KI2 motifs) provides a sufficient number of MELT repeats to generate an "in vivo-like" response. Generating a full-length version of the 2316-(isoform 2, used here) or 2342-residue (isoform 1) KLN1 protein, which is mostly intrinsically disordered, turned out to be impractical.

Specific phosphorylation of MAD1 and BUB1 by MPS1 is essential to promote efficient MCC formation in our MCC assembly assay. Yet, we have no evidence of robust MPS1 recruitment to our recombinant kinetochores. We are currently unable to say whether this is due to the very rapid kinetics of MPS1 recruitment to kinetochores, which may prevent formation of a stable complex in vitro, or to the absence of the appropriate binding sites. The characterization of the binding sites for MPS1 may be incomplete. In unpublished results (SG, VC, & AM), we have observed normal MPS1 recruitment in the presence of several Ndc80C mutants originally proposed to impair kinetochore recruitment of MPS1[43,44], possibly indicating that the determinants of this interaction remain to be identified.

Thus, an essential goal of our future studies will be to reproduce conditions for robust recruitment of all critical determinants of MCC assembly. Our in vitro reconstitution of the SAC monitors the time-dependence of MCC assembly in the absence of activities that disassemble MCC or that dephosphorylate the catalysts implicated in MCC assembly. Clearly, a fundamental function of kinetochores is to coordinate SAC signaling and microtubule binding, turning off MCC production upon biorientation. Such negative regulators include ATP-dependent, MCC-disassembly factors that counter the spontaneous assembly of MCC, phosphatases that remove crucial phosphorylation sites required for catalysis, and, in metazoans, molecular motors that disassemble the kinetochore corona upon microtubule binding[5,6]. The responsiveness of the SAC on kinetochore status depends on the function of these additional factors. Thus, another essential goal for future biochemical reconstitutions will be to include the negative regulators of MCC production.

## Methods
### Production of recombinant proteins

*Escherichia coli* BL21 (DE3) cells containing vectors expressing KNL1[M5] and its mutants were cultured in Terrific Broth at 37 °C until reaching an OD600 of 0.8–1. At this point, 0.3 mM IPTG was added, and the culture was further incubated at 18 °C for approximately 15 h. Cell pellets were then suspended in lysis buffer (50 mM Hepes pH 8, 300 mM NaCl, 5% glycerol, 2 mM TCEP) supplemented with a protease inhibitor cocktail (Serva). After sonication-induced lysis, the lysate was clarified by centrifugation at 90,000 × g at 4 °C for

1 h. The resulting lysate was filtered (0.8 µm) and applied onto Ni Sepharose® High Performance beads that had been pre-equilibrated in lysis buffer. Following a 2-hour incubation at 4 °C, the beads were washed with 50 volumes of lysis buffer and elution was performed with a buffer containing 300 mM imidazole. The eluted fractions containing the protein of interest were pooled, concentrated, and subjected to size exclusion chromatography (SEC). The SEC-purified fractions were concentrated, rapidly frozen in liquid nitrogen, and stored at −80 °C. *Escherichia coli* BL21 (DE3) cells containing vectors expressing KNL1[C] underwent a similar cultivation process in Terrific Broth. After induction with 0.1 mM IPTG and subsequent incubation at 18 °C for ~15 h, cell pellets were lysed, clarified and processed as described above. Cell pellets were suspended in lysis buffer (50 mM Tris pH 8, 50 mM NaCl, 5% glycerol, 2 mM TCEP). The cleared lysate was applied to Glutathione (GSH) beads, pre-equilibrated in lysis buffer and incubated at 4 °C for 4 h, allowing the GSH beads to selectively bind to the protein. The GSH beads were washed with lysis buffer and after extensive washing the protein was cleaved overnight with a GST-PreScission protease to remove the GST tag and gain an untagged protein. The cleaved protein was then collected from the GSH beads and was concentrated using centrifugal filters with a 10 kDa mass cut-off (Merck). The concentrated protein sample was further purified using a Superdex 200 16/600 size-exclusion column, pre-equilibrated in SEC buffer (50 mM Hepes pH 8.0, 50 mM NaCl, 5% glycerol, 2 mM TCEP). Other proteins were purified following detailed protocols[16,23,79,98,99]. Discussed mutants were as follows: ΔCM1 (Δ457-474), KEN1 (K535A, E536A, N537A), KEN2 (K625A, E626A, N627A), ABBA (F527A, F530A), KEN-ABBA (F527A, F530A, K535A, E536A, N537A), ΔHELIX (Δ271-409), KNL1-M1 (138-250, isoform 2), KNL1-M5A (MELT/MELA + SHT/AHA in MELT repeats 1, 12-15), KNL1-KI1 + 2/A (K176A, I177A, K212A, I213A, isoform 2).

### FRET measurements

The FRET sensors employed in this study were previously characterized in Faesen et al., 2017 and Piano et al., 2021. These sensors were utilized to investigate MCC assembly under various conditions. All measurements were conducted on a Clariostar plate-reader (BMG Labtech) using UV clear 96-well-plate flat-bottomed (Greiner). Data analysis and visualization were performed using Prism 9 software (GraphPad Software, Inc.). To monitor MCC assembly, the sensor components, along with MCC assembly catalysts, KMN, Rapamycin were combined in a final reaction volume of 100 µl, and the reading was initiated in buffer containing fresh 50 mM Hepes (pH 7.5), 150 mM NaCl, 5% glycerol, and 10 mM ß-Mercaptoethanol. Unless otherwise stated, assays utilized a final concentration of 100 nM for the FRET pair components ([CFP]BUBR1 and O-MAD2[TAMRA]) and were added the last before starting the measurements. The final concentration of catalysts varied

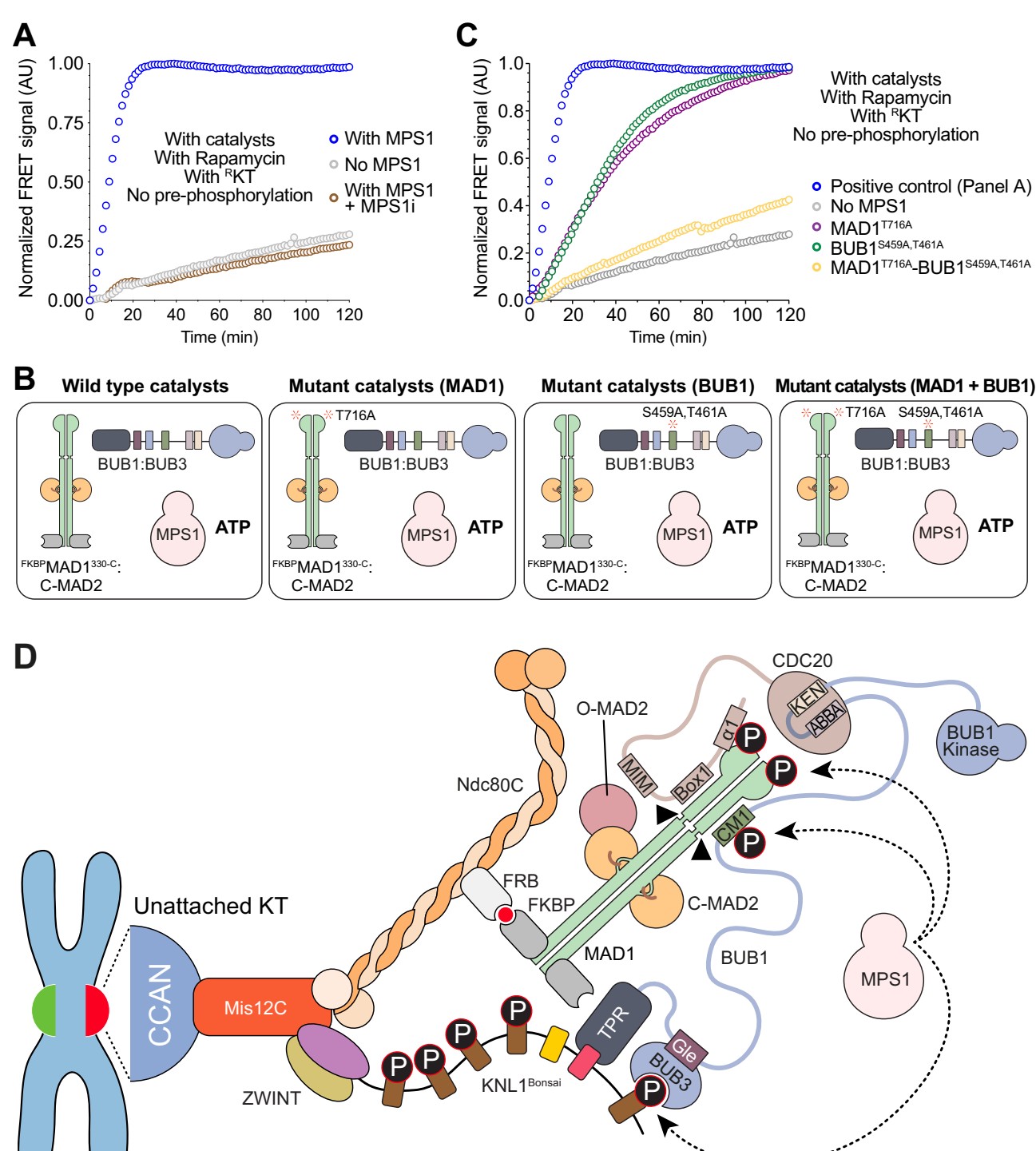

**Fig. 7 | BUB1 and MAD1:C-MAD2 are an integrated platform for catalytic MCC assembly by MPS1. A** MCC FRET assay monitoring the assembly of MCC without pre-phosphorylation of the components by MPS1, in the presence of the catalysts and Rapamycin, and with $^R$KT (blue curve, already shown in Fig. 6F), without MPS1 (gray curve, also shown in **C** and in the presence of the MPS1 inhibitor Reversine (brown curve). **B** Schematic representation of the experimental setup of the following assays. **C** MCC FRET assay monitoring the assembly of MCC without pre-phosphorylation of the components by MPS1, in the presence of Rapamycin and $^R$KT (blue curve, already shown in Fig. 6F), and with wild-type or mutant catalysts: BUB1$^{S459A-T461A}$ (green curve), MAD1$^{T716A}$ (purple curve), both BUB1$^{S459A-T461A}$ and MAD1$^{T716A}$ (yellow curve), or without MPS1 (gray curve, already shown in **A**. Panels reporting time-dependent changes in FRET signal are from single measurements and representative of at least three independent technical replicates of the experiment. **D**) Schematic representation of the assembly of the catalytic scaffold on the engineered outer kinetochore, highlighting the crucial binding interfaces and phosphorylation sites for efficient MCC formation.

depending on the specific experiment, which will be specified in the corresponding sections. Mixtures were excited using the filter 430-10 nm and the dichroic filter LP 504 nm and the emissions were scanned from 450 to 650 nm. Single wavelength acceptor fluorescence measurements were performed using the emission filter 590-20 nm, reading every 90 seconds and mixing 60 seconds at 500 rpm just before starting the measurements; focal height 6 mm, 200 flashes, gain 1200.

## Analytical size-exclusion chromatography

All proteins were diluted to a concentration of 3 µM in 50 µl buffer (50 mM Hepes pH 8.0, 5% glycerol, 150 mM NaCl, 2 mM TCEP) and incubated for 1 h at 4 °C before loading on Superose 6 5/150 column (GE Healthcare) equilibrated with the same buffer on an ÄKTA-micro system (GE Healthcare). Elution of proteins was monitored at 280 nm. Fractions (100 µl) were collected and analyzed by SDS–PAGE and Coomassie blue staining.

## Pull-down experiments

For pulldown experiments, KNL1$^C$ and KNL1$^{M5}$ were incubated together for -14 h at 4 °C with MPS1 kinase (1:10 kinase to substrate ratio) with 2 mM ATP, 10 mM MgCl$_2$ to get phosphorylated KNL1$^{Bonsai}$. To remove possible aggregates after the 14 h incubation, samples were spun at 21300 × $g$ for 30 minutes in a benchtop centrifuge (at 4 °C). Pre-equilibrated 10 µl dried GST beads for 1 h on ice with 2 µM in 40 µl of $^{GST}$CENP-C$^{1–71}$/$^{GST}$CENPC$^{1–544}$. The beads were then transferred to Pierce micro-spin columns (Thermo Fischer Scientific). The supernatant was then removed by centrifugation (5 min, 1000x$g$, 4 °C, same for all the following steps) and the remaining bait was added (KMN/CENP11/CENP11-KMN). After 1 h incubation on ice, the supernatant was again removed and washed once before adding the prey. Again, after 1 h incubation on ice, supernatant was removed. To remove the unbound material from the beads, beads were washed twice with 200 µl buffer (50 mM Hepes pH 8, 5% glycerol, 150 mM NaCl, 2 mM TCEP). The proteins were eluted in 25 µl buffer + 20 mM GSH pH 8, to which 5 µl 5x SDS-sample buffer were added.

## Cell culture

HeLa cells were grown in Dulbecco's Modified Eagle's Medium (DMEM; PAN Biotech) supplemented with 10% tetracycline-free FBS (PAN Biotech) and 2 mM L-Glutamine (PAN Biotech). Cells were grown at 37 °C with 5% CO$_2$.

## RNAi transfection

The following siRNAs oligos were used in this study: 60 nM siKNL1 (Invitrogen, HSS183683 5′-CACCCAGUGUCAUACAGCCAAUAUU-3′; HSS125942 5′-UCUACUGUGGUGGAGUUCUUGAUAA-3′; CCCUCUGG AGGAAUGGUCUAAUAAU-3′) for 24 h, 60 nM siZWINT (Sigma-Aldrich, 5′- GCACGUAGAGGCCAUCAAA-3′) for 48 h, 60 nM siNdc80C (Sigma-Aldrich, siNDC80 5′GAGUAGAACUAGAAUGUGA-3′; siSPC24 5′-GGA CACGACAGUCACAAUC-3′; siSPC25 5′-CUACAAGGAUUCCAUCAAA-3′) for 48 h. For Knl1C depletion, the protocol begins with siRNA oligos against ZWINT (reverse transfection), followed after 24 h by transfection of KNL1 oligos (forward transfection). For Ndc80C depletion, all siRNAs were added at the same time and the transfection was repeated after 24 h (first transfection reverse and second forward). siRNAs were transfected using RNAiMAX (Invitrogen) according to the manufacturer's instructions.

## Analysis of mitotic checkpoint functionality

10 000 HeLa cells were seeded in a 24-well imaging plate (ibidi GmbH) in DMEM and transfected with the respective siRNAs. After depletion, cells were synchronized in G2 phase with RO3306 (9 µM, Calbiochem). After 16 h the treatment was washed out four times with pre-warmed PBS and media was replaced with CO$_2$-independent L-15 imaging media supplemented with 10% FBS, 500 nM siR-DNA (Spirochrome) and 33 nM Nocodazole (Sigma-Aldrich). Cells were incubated for 6 h before imaging. Cells were imaged at 37 °C using a customized 3i Marianas system (spinning disk confocal) equipped with an Axio Observer Z1 microscope (Zeiss), a CSU-X1 confocal scanner unit (Yokogawa Electric Corporation, Tokyo, Japan), 40×/1.4 NA Oil Objective (Zeiss), and Orca Flash 4.0 sCMOS Camera (Hamamatsu). Images were acquired as z sections of 2 µm using Slidebook Software 6 (Intelligent Imaging Innovations). Manual count of mitotic cells was performed on maximal

intensity projections using the software Fiji (v. 2.0.0). Measurements were graphed with GraphPad Prism 10.2.

## Electroporation of recombinant proteins in living cells

For KNL1 electroporation experiments, recombinant GFP-labeled KNL1$^{Bonsai}$ was added to a final concentration of 6 µM in the electroporation slurry, as previously described[86]. For the electroporation of Ndc80C$^{FRB}$ or Ndc80C$^{FRB}$ /$^{FKBP}$MAD1$^{330-C}$:C-MAD2, the recombinant complexes were incubated at 4 °C for 25 min with or without Rapamycin (200 nM). After incubation Ndc80C$^{FRB}$ or Ndc80C$^{FRB}$ /$^{FKBP}$MAD1$^{330-C}$:C-MAD2 were added in the electroporation slurry at a final concentration of 4.5 µM. A Neon Transfection System (Thermo Fisher Scientific) was used. During and after the electroporation cells were treated with Rapamycin at a final concentration of 500 nM, when stated. Following an 8 h recovery lag, cells were treated with 9 µM RO3306 (Calbiochem) for 15 h. Subsequently, cells were released into mitosis in DMEM supplemented with of 3.3 µM Nocodazole (Sigma-Aldrich) or 10 µM MG132 (Calbiochem) for 1 h before fixation for immunofluorescence or harvesting for immunoblotting.

## Live imaging

After RO3306 synchronization a fraction of Ndc80C$^{FRB}$ /$^{FKBP}$MAD1$^{330-C}$:C-MAD2 electroporated cells was released in phenol red-free DMEM-F12 supplemented with 10% FBS, 500 nM siR-DNA (Spirochrome), with or without 500 nM Rapamycin. Cells were imaged at 37 °C for 18 h using a DeltaVision Elite System (GE Healthcare) equipped with an IX-71 inverted microscope (Olympu), a UPlanFLN 40 × 1.3 NA objective (Olympus) and a pco.edge sCMOS camera (PCO-TECH). Time-lapse movies were set up using the softWoRx software and acquired as z-scans (6 z-sections per field of view, 2 µm each) starting from the bottom of the plate, one channel at a time per z at 5 min intervals for 18 h. All images were visualized and processed using Fiji (v. 2.0.0). Prism 9 software (GraphPad Software, Inc.) was used for data analysis and visualization.

## Immunofluorescence

Cells were grown on coverslips pre-coated with Poly-L-lysine (Sigma-Aldrich). For IF experiments cells were pre-permeabilized with 0.5% Triton X-100 solution in PHEM (Pipes, HEPES, EGTA, MgSO$_4$) buffer supplemented with 100 nM microcystin for 5 mins and subsequently fixated with 4% PFA in PHEM for 20 mins. Afterwards cells were blocked for 1 h with 5% boiled goat serum (BGS) in PHEM buffer, and incubated for 2 h at room temperature with the following primary antibodies: CREST/anti-centromere antibodies (Antibodies, Inc., 1:200), anti-CENP-C (guinea pig polyclonal, MBL-PD030, MBL, 1:1000), anti-NDC80 (mouse, clone 9G3, Gene-Tex, Inc., 1:1000). Primary antibodies were diluted in 2.5% BGS-PHEM supplemented with 0.1% Triton X-100. Subsequently, cells were incubated for 1 h sat room temperature with the following secondary antibodies: anti-MAD1-DyLight550 (mouse, in-house generated, 1:500), goat anti-human Alexa Fluor 488 (Invitrogen, Carlsbad, California, USA, 1:200), goat anti-human Rodamine Red (Jackson Immunoresearch, 1:200). All washing steps were performed with PHEM-T buffer. DNA was stained with 0.5 µg/ml DAPI (Serva), and Mowiol (Calbiochem) was used as mounting media.

## Cell imaging

Cells were imaged at room temperature using a spinning-disk confocal device on the 3i Marianas system equipped with an Axio Observer Z1 microscope (Zeiss), a CSU-X1 confocal scanner unit (Yokogawa Electric Corporation), 100 × /1.4NA Oil Objectives (Zeiss), and Orca Flash 4.0 sCMOS Camera (Hamamatsu). Images were acquired as z sections at 0.27 µm (using Slidebook Software 6 from Intelligent Imaging Innovations or using LCS 3D software from Leica). Images were converted into maximal intensity projections, exported, and converted

into 16-bit TIFF files. Figures were arranged using Adobe Illustrator 2022 and 2025.

## Immunoblotting

Mitotic cells were collected through shake-off and resuspended in lysis buffer (150 mM KCl, 75 mM Hepes pH 7.5, 1.5 mM EGTA, 1.5 mM MgCl$_2$, 10% glycerol, and 0.075% NP-40 supplemented with protease inhibitor cocktail (Serva) and PhosSTOP phosphatase inhibitors (Roche). After lysis the whole-cell lysates were centrifuged at 21300 × $g$ for 30 mins at 4 °C. Afterwards, the supernatant was collected and resuspended in sample buffer for analysis by SDS-PAGE and Western blotting. The following primary antibodies were used: anti-GFP (rabbit, in-house, 1:1,000), anti-tubulin (mouse monoclonal, Sigma-Aldrich, 1:8,000), anti-NDC80 (mouse, clone 9G3, Gene-Tex, Inc., 1:1000), anti-MAD1 (mouse, in house made mouse monoclonal, clone BB3-8, 1:100), anti-GAPDH (rabbit, Sigma-Aldrich, 1:1000), anti-MBP (mouse, New England Biolabs, 1:10000), anti-BUB1 (rabbit, Abcam #9000, 1:2000), anti-KNL1pMELT (rabbit, 1:1000, kindly shared by the Kops group, Hubrecht Institute, Utrecht), anti-MAD1pThr716 (rabbit, 1:1000, kindly shared by the Saurin group, University of Dundee). As secondary antibodies, anti-mouse or anti-rabbit (NXA931 and NA934; Amersham, 1:5000) conjugated to horseradish peroxidase were used. After incubation with ECL Western blotting reagent (GE Healthcare), images were acquired with the ChemiDoc MP System (Bio-Rad) using Image Lab 6.0.1 software.

## Reporting summary

Further information on research design is available in the Nature Portfolio Reporting Summary linked to this article.

## Data availability

The data that support the findings of this study are included in the Source Data file, or available from the corresponding author upon request. Source data are provided with this paper.

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

## Acknowledgements

We thank Andrea Ciliberto for critical reading of the manuscript, all members of the Musacchio laboratory for helpful discussions, Amal Alex, Sara Carmignani, Carolin Körner, Sabine Wohlgemuth, Ingrid Hoffmann, Isidora Arias, and Javier Aviles for help with reagents preparation. A.M. acknowledges funding from the Max Planck Society, the European Research Council (ERC) Synergy Grant 951430 (BIOMECA-NET), the DGF's Collaborative Research Center 1430 "Molecular Mechanisms of Cell State Transitions", and the CANTAR network under the Netzwerke-NRW program.

## Author contributions

Conceptualization: S.S., V.P., S.G., A.M. Investigation: S.S., S.G., V.P., V.C., B.G. Funding acquisition: A.M. Project Administration: A.M. Resources: P.S. Supervision: A.M. Validation: S.S., S.G., V.P., A.M. Visualization: S.S., S.G., A.M. Writing—original draft: A.M. Writing—review & editing: All authors.

## Funding

## Competing interests

The authors declare no competing interests.
