## [Transparent Peer Review file · Nature Communications]

Interplay of kinetochores and catalysts drives rapid assembly of the mitotic checkpoint complex

Corresponding Author: Professor Andrea Musacchio

Version 0:

Reviewer comments:

Reviewer #1

(Remarks to the Author)

This manuscript reconstitutes the spindle assembly checkpoint (SAC) in vitro focusing on the role of kinetochores in stimulating MCC production. The SAC is a fundamental signaling mechanism ensuring proper segregation of the genetic material during mitosis. Work over several decades have identified the components of the SAC and detailed mechanistic insight has been obtained from both cell-based assays as well as biochemical reconstitutions pioneered by the lab. A holy grail has been to reconstitute SAC signaling and MCC production in vitro which the lab previously did but without kinetochores. Here the authors expand on their previous in vitro system by including kinetochores and showing that this stimulates MCC production – an important result for the field. Collectively the work is of very high quality and sets a new bar for what can be reconstituted in vitro and will be of interest to both people in the field but also biochemists working on reconstituting biological systems. For these reasons I strongly support its publication in NCOMMS but have a few points that I think would strengthen the manuscript and make it more impactful.

Major points:

- 1) For technical reasons the authors resort to a system of chemical induction of the Ndc80-Mad1/2 interaction as well as an engineered KNL1 construct. I think it would be important to show that these engineered systems actual are functional in vivo (I suspect they will be). I suggest the authors use their depletion strategy (Fig. 1C) and combine it with their ability to transfect proteins into cells to test this.
- 2) I am struggling with their conclusion that the kinetochore does not simply act by concentrating SAC components based on the fact that the CM1 motif of Bub1 is still needed for MCC production in their system. I think the literature is clear on showing the importance of the Mad1-Bub1 interaction for SAC signalling but this in my view does not rule out that the function of the kinetochore could not be to concentrate (rather than specific positioning) these proteins to facilitate their binding. Could the authors be more specific in the discussion on what experiments in the presented work makes them able to discriminate between a function in concentrating proteins versus specific positioning. In my view this would require more engineering work to show if the position of the Bub1-Bub3 binding sites relative to Mad1/2 is important which have not been done.
- 3) A fundamental question is whether microtubules would prevent the stimulation of MCC production by KT's in their system. I suggest the authors test this if possible.
- 4) Is the binding of the MCC to APC/C-Cdc20 affected by KT's in their system?

Minor points:

- 1) Can the authors comment on the stoichiometry of Mps1 phosphorylation sites in their assays?
- 2) Page 2 line 60: SAC should be MCC
- 3) Page 4 line 130: For the RNAi depletions please specify what is being depleted (pool of oligoes targeting Ndc80C of KNL1C components?).
- 4) Page 5 line 163-164: please rephrase to improve reading.
- 5) Page 7 line 217-218: they cannot make the claim that they have a system for initiating SAC signalling only that they have a system for recruitment. Please rephrase.
- 6) Page 11 line 359-360: I would make clear that this claim is likely only relevant in their in vitro system. CM1 seems very essential even in systems were Mad1 is tethered (Heinrich et al 2014).

7) Information of number of repeats of experiments is missing in figure legends.

Reviewer #2

(Remarks to the Author)

The spindle assembly checkpoint (SAC) is an essential cell cycle checkpoint that coordinates anaphase onset with the correct attachment of kinetochores to microtubules. The SAC is exerted by the mitotic checkpoint complex (MCC) a complex of four proteins whose intrinsic assembly rate is extremely slow, but that is accelerated by unattached kinetochores. To address how unattached kinetochores catalyse MCC assembly, the Musacchio group and others have previously discovered that a phosphorylation cascade initiated by the MPS1 protein kinase regulates the association of MCC assembly catalysts to promote MCC assembly by catalysing the conversion of open-MAD2 to closed MAD2.

In this manuscript the authors address the question of how kinetochores catalyse the assembly of the MCC. The study builds on their previous work developing an elegant FRET-based assay to measure rates of MCC assembly, and their advances in reconstituting kinetochores in vitro. Here they reconstitute the outer kinetochore KMN network complex and test its effect on rates of MCC assembly in the presence of MCC assembly catalysts. The major findings are the demonstration that the reconstituted kinetochore enhances the assembly of the MCC mediated by MCC catalysts when both MAD1:MAD2 and BUB1:BUB3 are recruited to the recombinant KMN network complex. The former was recruited artificially by means of Rapamycin/FRB/FKBP whereas BUB1:BUB3 was recruited to the KNL1 through the KI1, KI2 and phosphorylated MELT motifs of KNL1. Using this system the authors test the role of various motifs in KNL1 and BUB1. The authors show that the MELT and KI1 and KI2 motifs of KNL1 and ABBA, KEN and CM1 motifs in BUB1, and its phosphorylation, are required for optimal rates of MCC assembly.

Overall this is an important and interesting paper that advances our understanding of the role and mechanism of kinetochore-mediated enhancement of rates of MCC assembly, consistent with the idea that efficient rates of MCC assembly and SAC signalling depend on proximity (and likely correct orientation) of BUB1:BUB3 and MAD1:MAD2 in a catalytic scaffold that binds CDC20. Given that the authors recruited MAD1:MAD2 to the outer kinetochore using artificial dimerisation, as the authors indicate, it will be interesting for future work to understand the basis of MAD1:MAD2 recruitment to the outer kinetochore.

1. Figure 1C. Show p values.

2. Label O-MAD2 and CDC20 in Fig. 3A.

3. Fig. 3B: In this experiment with recombinant KT_s, MAD1-MAD2 is recruited to the complex through rapamycin and NDC80C. Another path might be through BUB1-BUB3 binding directly to MAD1. Did the authors compare rates of MCC assembly with and without rapamycin?

4. Define CENP12 in this paper.

5. Figures 4 & 5. The authors test the role of the BUB1 KEN, ABBA and CM1 motifs in enhancing rates of MCC assembly. Mutating both the KEN and ABBA motifs caused a dramatic decrease in MCC assembly rates. Did the authors test the roles of the KEN and ABBA motifs individually? It would be worthwhile to do so.

6. Unlike the KEN and ABBA motif mutants, mutating CM1 caused a smaller reduction of MCC assembly rates. In this experiment it wasn't clear whether rapamycin was present or not. Moreover since the affinity of MAD1 for CM1 is enhanced by CM1 phosphorylation, the authors should show how the rates of MCC assembly are affected when CM1 is deleted in the context of pre-phosphorylated BUB1. One might expect a more substantial reduction in the rate of MCC assembly when CM1 is deleted when the CM1 motif is phosphorylated.

7. What was the concentration of catalysts used in the experiments shown in Fig 4 and Fig. 5. The authors state that this was high. Could the effects of mutating the BUB1 motifs and lack of phosphorylation be amplified at lower catalyst concentrations (i.e. 20 nM MAD1-MAD2 and 40 nM BUB1-BUB3 as used in Fig. 3)?

8. Figure 5C. The reduction in rate of MCC assembly for the MAD1-T716A mutant and the BUB1-S459A,T461A mutant relative to the control seem similar. In the text the authors state that the BUB1-S459A,T461A mutant 'caused a moderate decline in the rate of MCC assembly' whereas the MAD1-T716A mutant caused 'a pronounced decrease of the rate of MCC assembly'. To this reviewer the description of these mutants on the rate of MCC assembly doesn't match the results in Fig. 5C.

Minor comments

1. Lines 162-163. Sentence is difficult to understand.

2. Lines 170-171. The authors state that so far they have been unable to identify a minimal set of interactions leading to robust MAD1:MAD2 recruitment to reconstituted kinetochores in vitro. Can the authors state what has been attempted (perhaps in Supplementary Information).

3. Line 235. 'was' is missing

Version 1:

Reviewer comments:

Reviewer #1

(Remarks to the Author)

The authors have addressed all my concerns with new experiments or through textual changes. I recommend publication of this nice work.

Reviewer #2

(Remarks to the Author)

The authors have addressed my questions and comments. This interesting manuscript warrants publication in Nature Communications.

REVIEWER COMMENTS

We thank the reviewers for their very constructive comments on our manuscript. We also apologize for the long delay in resubmitting the revised manuscript. The delay was caused by the already planned departure of laboratory members and the need to reorganize tasks to complete the revision. We have now revised the manuscript quite extensively, adding several new experiments and control experiments in response to the reviewers' question. In a few cases where we felt that the tools in our hands did not yet allow us to provide rigorous answers to the reviewers' comments, we preferred textual answers to providing rushed experiments. These few cases are thoroughly discussed below.

Reviewer #1 (Remarks to the Author):

This manuscript reconstitutes the spindle assembly checkpoint (SAC) *in vitro* focusing on the role of kinetochores in stimulating MCC production. The SAC is a fundamental signaling mechanism ensuring proper segregation of the genetic material during mitosis. Work over several decades have identified the components of the SAC and detailed mechanistic insight has been obtained from both cell-based assays as well as biochemical reconstitutions pioneered by the lab. A holy grail has been to reconstitute SAC signaling and MCC production *in vitro* which the lab previously did but without kinetochores. Here the authors expand on their previous *in vitro* system by including kinetochores and showing that this stimulates MCC production – an important result for the field. Collectively the work is of very high quality and sets a new bar for what can be reconstituted *in vitro* and will be of interest to both people in the field but also biochemists working on reconstituting biological systems. For these reasons I strongly support its publication in NCOMMS but have a few points that I think would strengthen the manuscript and make it more impactful.

We are very grateful to the reviewer for his/her support of our work and for many insightful comments.

Major points:

1) For technical reasons the authors resort to a system of chemical induction of the Ndc80-Mad1/2 interaction as well as an engineered KNL1 construct. I think it would be important to show that these engineered systems actual are functional *in vivo* (I suspect they will be). I suggest the authors use their depletion strategy (Fig. 1C) and combine it with their ability to transfect proteins into cells to test this.

We thank the reviewer for this suggestion. We incorporated a new functional analysis of Ndc80^{FRB} and ^{FKBP}MAD1^{330-C}:MAD2 electroporated in HeLa cells depleted of Ndc80C, and presented it in a newly organized Figure 2. The analysis demonstrates that Ndc80^{FRB} recruits ^{FKBP}MAD1^{330-C}:MAD2 and that the latter maintains the checkpoint in metaphase-arrested cells in presence of Rapamycin. We also detected a mitotic delay with Ndc80^{FRB}, probably due to its suboptimal levels or to an imperceptible alignment error. Collectively, these new data demonstrate that the Ndc80^{FRB} and ^{FKBP}MAD1^{330-C}:MAD2 system is functional. We also tried to extend our results in Figure S1C-D (already demonstrating kinetochore localization of KNL1^{Bonsai}) to show that KNL1^{Bonsai} is also checkpoint proficient. Despite repeated attempts, we were unable to observe robust kinetochore localization of electroporated KNL1^{Bonsai} in cells that had been previously depleted of KNL1. We have not yet identified an obvious reason for why KNL1^{Bonsai} appears to be less stable in absence of endogenous KNL1.

2) I am struggling with their conclusion that the kinetochore does not simply act by concentrating SAC components based on the fact that the CM1 motif of Bub1 is still needed for MCC production in their system. I think the literature is clear on showing the importance of the Mad1-Bub1 interaction for SAC signalling but this in my view does not rule out that the function of the kinetochore could not be to concentrate (rather than specific positioning) these proteins to facilitate their binding. Could the authors be more specific in the discussion on what experiments in the presented work makes them able to discriminate between a function in concentrating proteins versus specific positioning. In my view this would require more engineering work to show if the position of the Bub1-Bub3 binding sites relative to Mad1/2 is important which have not been done.

We thank the reviewer for raising this point. To be sure, we already write in the Abstract that “Our observations depict kinetochores as a cradle that catalyzes rapid MCC assembly by concentrating and co-orienting distinct SAC catalysts.” A similar statement is re-iterated at the end of the Introduction. In other words, we agree that concentration plays an important role. In this revised version of the manuscript, we have incorporated the reviewer’s concern and clarified our points even further. Specifically, we included ‘possibly’ in the abstract (“...by concentrating and *possibly* co-orienting distinct SAC catalysts”). We also added the following sentence to the Conclusions: “Nonetheless, we cannot exclude that the sole function of kinetochores is simply to recruit the SAC’s catalytic components, without major restraints on their reciprocal positioning.”

3) A fundamental question is whether microtubules would prevent the stimulation of MCC production by KT’s in their system. I suggest the authors test this if possible.

We agree with the reviewer that this is a crucial question. Regretfully, we also feel that we are not yet in a position to answer it. Briefly, we have now gathered biochemical evidence that MPS1 kinase does not compete with microtubules for kinetochore recruitment, in line with work from the Gruneberg and Petronczki laboratories. The regulation of MPS1 activity at kinetochores is strictly dependent on Aurora B kinase, but our experiments *in vitro* do not clarify why this is the case. In other words, we have not yet reconstituted a faithful microtubule-dependent switch (which controls Aurora B kinase activity). That makes us reluctant adding microtubules, as we feel that any effect on the rate of MCC assembly would be hardly interpretable given the current evidence. We hope the reviewer will agree with us.

4) Is the binding of the MCC to APC/C-Cdc20 affected by KT’s in their system?

Also in this case, we definitely appreciate the importance of this question, but also feel that answering it would go beyond the scope of the present manuscript. The crucial reason for this is that we do not have a direct, real-time sensor to measure the MCC-APC/C interaction concomitantly with MCC assembly and, ideally, APC/C inhibition. While we plan to include precisely this type of experiment in future work, we are not in a position to perform these experiments effectively at this time.

Minor points:

1) Can the authors comment on the stoichiometry of Mps1 phosphorylation sites in their assays?

We have now added a panel (Figure S1I) that reports a phosphorylation analysis in presence of ³²P-KT and catalysts. While not directly revealing the stoichiometry of phosphorylation, this analysis shows progressive and eventually plateauing phosphorylation for a few critical substrates.

2) Page 2 line 60: SAC should be MCC

Thank you, we have corrected this.

3) Page 4 line 130: For the RNAi depletions please specify what is being depleted (pool of oligos targeting Ndc80C of KNL1C components?).

Yes, it was pools of oligos. We have now included a short description of the strategy used in the main text and have extended the description in Methods to clarify this.

4) Page 5 line 163-164: please rephrase to improve reading.

Thank you, the sentence has been rephrased.

5) Page 7 line 217-218: they cannot make the claim that they have a system for initiating SAC signalling only that they have a system for recruitment. Please rephrase.

We have changed the text accordingly. We now write that "...we have obtained an inducible system for recruiting SAC signaling components on reconstituted kinetochores."

6) Page 11 line 359-360: I would make clear that this claim is likely only relevant in their *in vitro* system. CM1 seems very essential even in systems were Mad1 is tethered (Heinrich et al 2014).

We agree with the reviewer. The old text was already indicating that this denotes a discrepancy, but we have now reinforced the point and write: "*In vivo*, inhibition of the BUB1:MAD1 binding interaction is incompatible with robust SAC signaling (Ballister et al., 2014; Heinrich et al., 2014; London and Biggins, 2014; Qian et al., 2017; Silio et al., 2015). Our observation that MCC assembly *in vitro* is only partially impaired when MAD1 and BUB1 cannot interact suggests that kinetochores, by recruiting BUB1 and MAD1:MAD2 in close proximity, compensate for the loss of an interaction of the catalysts. Nonetheless, the discrepancy also suggests that our system does not yet fully recapitulate all properties of SAC catalysis.

7) Information of number of repeats of experiments is missing in figure legends.

Thank you. The number of repeats of experiments has now been included to the legends.

Reviewer #2 (Remarks to the Author):

The spindle assembly checkpoint (SAC) is an essential cell cycle checkpoint that coordinates anaphase onset with the correct attachment of kinetochores to microtubules. The SAC is exerted by the mitotic checkpoint complex (MCC) a complex of four proteins whose intrinsic assembly rate is extremely

slow, but that is accelerated by unattached kinetochores. To address how unattached kinetochores catalyse MCC assembly, the Musacchio group and others have previously discovered that a phosphorylation cascade initiated by the MPS1 protein kinase regulates the association of MCC assembly catalysts to promote MCC assembly by catalysing the conversion of open-MAD2 to closed MAD2.

In this manuscript the authors address the question of how kinetochores catalyse the assembly of the MCC. The study builds on their previous work developing an elegant FRET-based assay to measure rates of MCC assembly, and their advances in reconstituting kinetochores *in vitro*. Here they reconstitute the outer kinetochore KMN network complex and test its effect on rates of MCC assembly in the presence of MCC assembly catalysts. The major findings are the demonstration that the reconstituted kinetochore enhances the assembly of the MCC mediated by MCC catalysts when both MAD1:MAD2 and BUB1:BUB3 are recruited to the recombinant KMN network complex. The former was recruited artificially by means of Rapamycin/FRB/FKBP whereas BUB1:BUB3 was recruited to the KNL1 through the KI1, KI2 and phosphorylated MELT motifs of KNL1. Using this system the authors test the role of various motifs in KNL1 and BUB1. The authors show that the MELT and KI1 and KI2 motifs of KNL1 and ABBA, KEN and CM1 motifs in BUB1, and its phosphorylation, are required for optimal rates of MCC assembly.

Overall this is an important and interesting paper that advances our understanding of the role and mechanism of kinetochore-mediated enhancement of rates of MCC assembly, consistent with the idea that efficient rates of MCC assembly and SAC signalling depend on proximity (and likely correct orientation) of BUB1:BUB3 and MAD1:MAD2 in a catalytic scaffold that binds CDC20. Given that the authors recruited MAD1:MAD2 to the outer kinetochore using artificial dimerisation, as the authors indicate, it will be interesting for future work to understand the basis of MAD1:MAD2 recruitment to the outer kinetochore.

We are very grateful to the reviewer for an encouraging assessment of our work.

1. Figure 1C. Show p values.

We have now included p values as requested by the reviewer.

2. Label O-MAD2 and CDC20 in Fig. 3A.

Thank you, we have now labelled the indicated proteins.

3. Fig. 3B: In this experiment with recombinant KT's, MAD1-MAD2 is recruited to the complex through rapamycin and NDC80C. Another path might be through BUB1-BUB3 binding directly to MAD1. Did the authors compare rates of MCC assembly with and without rapamycin?

Thank you for this comment. The control discussed by the reviewer was already present in the original manuscript, and is displayed in Figure 4D and Figure S2A.

4. Define CENP12 in this paper.

Thank you, we have defined CENP-12.

5. Figures 4 & 5. The authors test the role of the BUB1 KEN, ABBA and CM1 motifs in enhancing rates of MCC assembly. Mutating both the KEN and ABBA motifs caused a dramatic decrease in MCC assembly rates. Did the authors test the roles of the KEN and ABBA motifs individually? It would be worthwhile to do so.

We agree with the reviewer and we have now carried out new experiments with individual KEN1, KEN2, and ABBA mutants, and included them as panels E-G in Figure S4.

6. Unlike the KEN and ABBA motif mutants, mutating CM1 caused a smaller reduction of MCC assembly rates. In this experiment it wasn't clear whether rapamycin was present or not. Moreover since the affinity of MAD1 for CM1 is enhanced by CM1 phosphorylation, the authors should show how the rates of MCC assembly are affected when CM1 is deleted in the context of pre-phosphorylated BUB1. One might expect a more substantial reduction in the rate of MCC assembly when CM1 is deleted when the CM1 motif is phosphorylated.

Thank you for this comment. The presence of Rapamycin in these experiments is indicated by the text above the bar over panels D-F in Figure 6. Regarding the second part of the comment, unfortunately we cannot test this because the Δ CM1 construct (discussed in Piano et al. 2021) does not contain the phosphorylation sites. We note that phosphorylation of BUB1 at residue 461 is expected to enhance binding to MAD1.

7. What was the concentration of catalysts used in the experiments shown in Fig 4 and Fig. 5. The authors state that this was high. Could the effects of mutating the BUB1 motifs and lack of phosphorylation be amplified at lower catalyst concentrations (i.e. 20 nM MAD1-MAD2 and 40 nM BUB1-BUB3 as used in Fig. 3)?

We have collected the reaction conditions in Table 1. Experiments in former figures 4 and 5 (now 5 and 6) were carried out at the same low concentrations of catalysts used in the previous figures (20 nM MAD1:MAD2 and 40 nM BUB1:BUB3 used in previous Figure 3 (now 4)). Thus, the results are directly comparable.

8. Figure 5C. The reduction in rate of MCC assembly for the MAD1-T716A mutant and the BUB1-S459A,T461A mutant relative to the control seem similar. In the text the authors state that the BUB1-S459A,T461A mutant 'caused a moderate decline in the rate of MCC assembly' whereas the MAD1-T716A mutant caused 'a pronounced decrease of the rate of MCC assembly'. To this reviewer the description of these mutants on the rate of MCC assembly doesn't match the results in Fig. 5C.

We agree with the reviewer that the description of these experiments was confusing. We have rewritten the description and now write: "We combined mutations of Ser459 and Thr461 in a single mutant and performed the MCC assembly reaction in the presence of kinetochores. We observed a moderate decline in the rate of MCC assembly (Figure 7B-C). Additionally, when we replaced wild-type MAD1 with the MAD1^{T716A} mutant in the MCC assembly assay, we also observed a moderate decrease of the rate of MCC assembly (Figure 7B-C). When the phospho-alanine mutants MAD1^{T716A} and BUB1^{S459A-T461A} were combined, the rate of MCC assembly dropped substantially, becoming similar to the rate in the absence of MPS1 kinase (Figure 7B-C).

Minor comments

1. Lines 162-163. Sentence is difficult to understand.

Thank you, we corrected the sentence.

2. Lines 170-171. The authors that that so far they have been unable to identify a minimal set of interactions leading to robust MAD1:MAD2 recruitment to reconstituted kinetochores in vitro. Can the authors state what has been attempted (perhaps in Supplementary Information).

We were referring to the same conditions that trigger binding of ^{FKPB}MAD1^{330-C}:MAD2 to KMN immobilized on solid phase. To clarify this, we have now added the following sentence after the description of binding conditions for the ^{FKPB}MAD1^{330-C}:MAD2 to KMN: “Under the same conditions, we failed to observe binding of MAD1:MAD2 (SS and AM, unpublished observations), indicating that its robust recruitment necessitates of additional determinants, probably in the kinetochore corona.”

3. Line 235. ‘was’ is missing

Thank you, this has been corrected.